# Therapeutic inhibition of keratinocyte TRPV3 sensory channel by local anesthetic dyclonine

Qiang Liu[1†], Jin Wang[2†], Xin Wei[1], Juan Hu[1], Conghui Ping[1], Yue Gao[1], Chang Xie[1], Peiyu Wang[1], Peng Cao[3], Zhengyu Cao[4], Ye Yu[2], Dongdong Li[5], Jing Yao[1]*

[1]State Key Laboratory of Virology, Hubei Key Laboratory of Cell Homeostasis, College of Life Sciences, Frontier Science Center for Immunology and Metabolism, Wuhan University, Wuhan, China; [2]School of Basic Medicine and Clinical Pharmacy, China Pharmaceutical University, Nanjing, China; [3]Hospital of Integrated Traditional Chinese and Western Medicine, Nanjing University of Chinese Medicine, Nanjing, China; [4]State Key Laboratory of Natural Medicines and Jiangsu Provincial Key Laboratory for TCM Evaluation and Translational Development, School of Traditional Chinese Pharmacy, China Pharmaceutical University, Nanjing, China; [5]Sorbonne Université, Institute of Biology Paris Seine, Neuroscience Paris Seine, CNRS UMR8246, Inserm U1130, Paris, France

**\*For correspondence:**
jyao@whu.edu.cn

[†]These authors contributed equally to this work

**Competing interest:** The authors declare that no competing interests exist.

**Abstract** The multimodal sensory channel transient receptor potential vanilloid-3 (TRPV3) is expressed in epidermal keratinocytes and implicated in chronic pruritus, allergy, and inflammation-related skin disorders. Gain-of-function mutations of TRPV3 cause hair growth disorders in mice and Olmsted syndrome in humans. Nevertheless, whether and how TRPV3 could be therapeutically targeted remains to be elucidated. We here report that mouse and human TRPV3 channel is targeted by the clinical medication dyclonine that exerts a potent inhibitory effect. Accordingly, dyclonine rescued cell death caused by gain-of-function TRPV3 mutations and suppressed pruritus symptoms in vivo in mouse model. At the single-channel level, dyclonine inhibited TRPV3 open probability but not the unitary conductance. By molecular simulations and mutagenesis, we further uncovered key residues in TRPV3 pore region that could toggle the inhibitory efficiency of dyclonine. The functional and mechanistic insights obtained on dyclonine-TRPV3 interaction will help to conceive therapeutics for skin inflammation.

## Introduction

Transient receptor potential (TRP) channels belong to a family of calcium-permeable and nonselective cation channels, essential for body sensory processing and local inflammatory development (*Clapham, 2003*). As a polymodal cellular sensor, transient receptor potential vanilloid-3 (TRPV3) channel is abundantly expressed in skin keratinocytes (*Chung et al., 2004b*; *Peier et al., 2002*; *Xu et al., 2002*) and in cells surrounding the hair follicles (*Cheng et al., 2010*). TRPV3 integrates a wide spectrum of physical and chemical stimuli (*Luo and Hu, 2014*). TRPV3 is sensitive to innocuous temperatures above 30–33°C and exhibits an increased response at noxious temperature (*Chung et al., 2005*; *Xu et al., 2002*). Natural plant products such as camphor (*Moqrich et al., 2005*), carvacrol, eugenol, thymol (*Xu et al., 2006*), and the pharmacological compound 2-aminoethoxydiphenyl borate (2-APB) (*Chung et al., 2004a*; *Colton and Zhu, 2007*) also activate TRPV3. In addition, TRPV3 is directly activated by acidic pH from cytoplasmic side (*Gao et al., 2016*).

Mounting evidence implicates TRPV3 channel in cutaneous sensation including thermal sensation (*Chung et al., 2004b*), nociception (*Huang et al., 2008*), and itch (*Yamamoto-Kasai et al., 2012*). They also participate in the maintenance of skin barrier, hair growth (*Cheng et al., 2010*), and wound healing (*Aijima et al., 2014*; *Yamada et al., 2010*). The dysfunction of TRPV3 channels has come to the fore as a key regulator of physiological and pathological responses of skin (*Ho and Lee, 2015*). In rodents, the Gly573Ser substitution in TRPV3 renders the channel spontaneously active and caused a hairless phenotype in DS-Nh mice and WBN/Kob-Ht rats (*Asakawa et al., 2006*). DS-Nh mice also develop severe scratching behavior and pruritic dermatitis. TRPV3 dysfunction caused by genetic gain-of-function mutations or pharmaceutical activation has been linked to human skin diseases, including genodermatosis known as Olmsted syndrome (*Agarwala et al., 2015*; *Lin et al., 2012*) and erythromelalgia (*Duchatelet et al., 2014*). Furthermore, TRPV3-deficient mice give rise to phenotypes of curly whiskers and wavy hair coat (*Cheng et al., 2010*). Conversely, hyperactive TRPV3 channels expressed in human outer root sheath keratinocytes inhibit hair growth (*Borbíró et al., 2011*). While being implicated in a variety of skin disorders, whether and how TRPV3 could be therapeutically targeted remains to be elucidated. It is thus desirable to identify and understand the clinical medications that potentially target TRPV3 channels.

Dyclonine is a clinical anesthetic characterized by rapid onset of effect, lack of systemic toxicity, and low index of sensitization (*Florestano and Bahler, 1956*). Its topical application (0.5% or 1% dyclonine hydrochloride contained in the topical solution, i.e., ~30.7 mM at a dose of 1%, according to the United States Pharmacopeia) rapidly relieves itching and pain in patients by ameliorating inflamed, excoriated, and broken lesions on mucous membranes and skin (*Morginson et al., 1956*). Accordingly, dyclonine is used to anesthetize mucous membranes prior to endoscopy (*Formaker et al., 1998*). The clinical scenario targeted by dyclonine treatment echoes the pathological aspects of TRPV3-related skin disorders, suggesting that the therapeutic effects of dyclonine might involve its interaction with TRPV3 sensory channel.

We report here that mouse and human TRPV3 (hTRPV3) channel activity was potently suppressed by dyclonine. It dose-dependently inhibited TRPV3 currents in a voltage-independent manner and rescued cell death caused by TRPV3 gain-of-function mutation. In vivo, dyclonine indeed suppressed the itching/scratching behaviors induced by TRPV3 channel agonist carvacrol as evidenced by the TRPV3 knock out (KO) mice. At single-channel level, dyclonine reduced TRPV3 channel open probability without altering the unitary conductance. We also identified molecular residues that were capable of either eliminating or enhancing the inhibitory effect of dyclonine. These data demonstrate the effective inhibition of TRPV3 channel by dyclonine, supplementing a molecular mechanism for its clinical effects and raising its potential to ameliorate TRPV3-associated disorders.

## Results

### Inhibition of TRPV3 currents by dyclonine

We first examined the effect of dyclonine on TRPV3 activity induced by the TRPV channel agonist 2-APB (100 μM). Whole-cell currents were recorded at a holding potential of –60 mV in HEK 293T cells expressing mouse TRPV3. Because TRPV3 channels exhibit sensitizing properties upon repeated stimulation (*Chung et al., 2004a*), we examined the effect of dyclonine after the response had stabilized following repetitive application of 2-APB (*Figure 1A*). The presence of 5 and 10 μM dyclonine significantly inhibited TRPV3 currents response to 30% ± 2% and 15% ± 3% of control level, respectively. After washing out of dyclonine, 2-APB evoked a similar response to the control level, indicating that the blocking effect of dyclonine is reversible (*Figure 1A,B*). We repeated the experiments with different doses of dyclonine. The dose-response curve indicates that dyclonine inhibited TRPV3 currents in a concentration-dependent manner with an $IC_{50}$ of 3.2 ± 0.24 μM (n = 6, *Figure 1C*). We further examined the inhibitory effect of dyclonine on TRPV3 activated by varying concentrations of 2-APB (*Figure 1D*). The dose-response curves to 2-APB were fitted with a Hill equation. The inhibitory effect of dyclonine on TRPV3 activation was consistently observed under all tested 2-APB concentrations (*Figure 1E*). The corresponding $EC_{50}$ values and Hill coefficients were not changed by the presence of dyclonine (*Figure 1E*, $EC_{50}$ = 22.93 ± 0.02 μM, $n_H$ = 1.6 ± 0.1 without dyclonine vs. $EC_{50}$ = 22.03 ± 0.86 μM, $n_H$ = 1.7 ± 0.1 with 3 μM dyclonine), as confirmed by the normalized dose-response

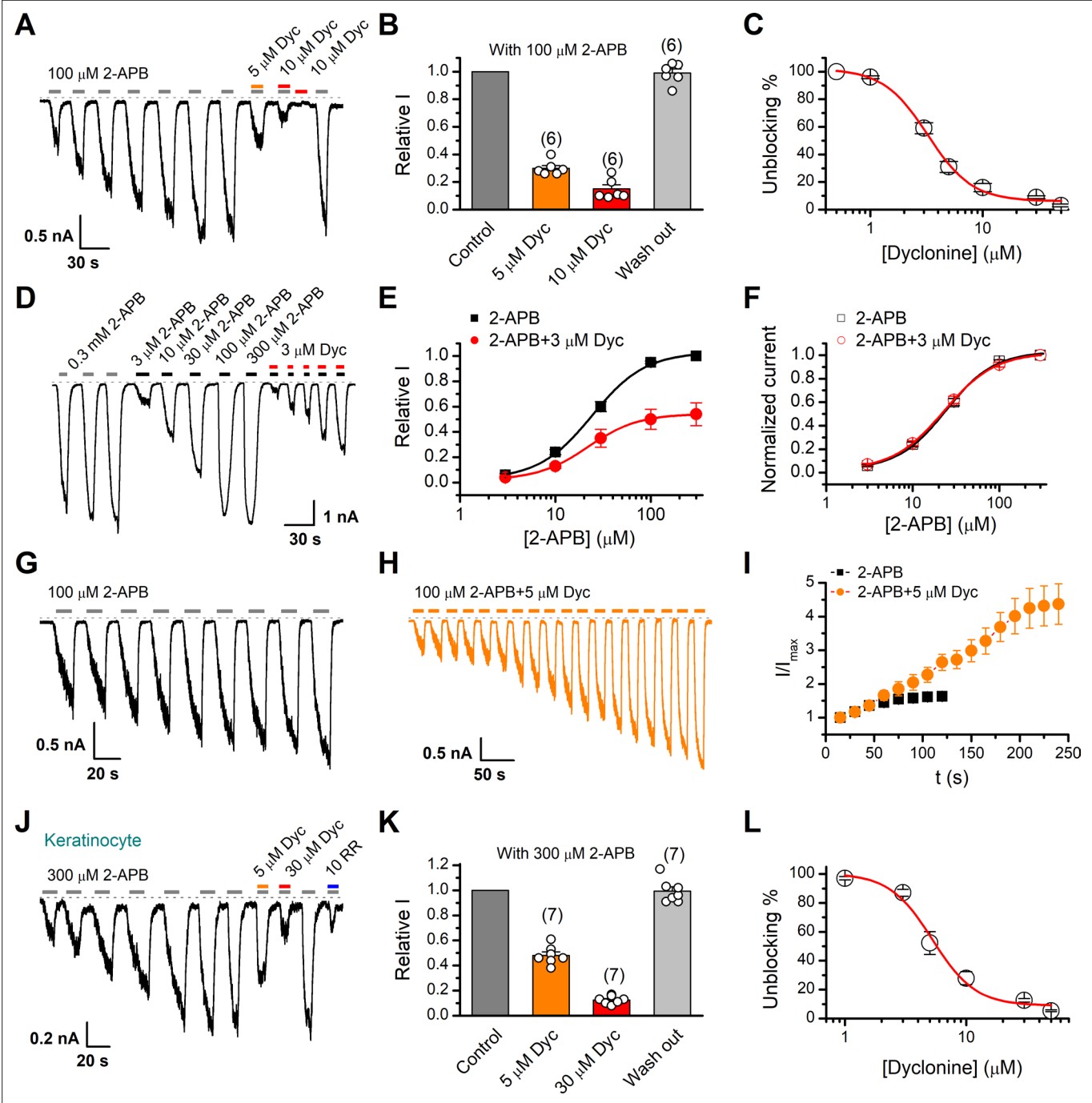

**Figure 1.** Inhibition of transient receptor potential vanilloid-3 (TRPV3) currents by dyclonine (Dyc). (**A**) Inhibition of 2-aminoethoxydiphenylborate (2-APB)-evoked currents by Dyc in a representative HEK 293T cell expressing mouse TRPV3. After sensitization by repeated application of 100 µM 2-APB, the cell was exposed to 5 or 10 µM Dyc with 100 µM 2-APB or 10 µM Dyc only as indicated by the bars. Membrane currents were recorded in whole-cell configuration, and the holding potential was –60 mV. (**B**) Summary of relative currents elicited by 100 µM 2-APB in the presence of 0, 5, or 10 µM Dyc. Numbers of cells are indicated in parentheses. (**C**) The dose-response curve for Dyc inhibition of TRPV3 currents was fitted by Hill equation (IC$_{50}$ = 3.2 ± 0.24 µM and $n_H$ = 2.2 ± 0.32, n = 6). (**D**) Representative whole-cell current traces showing the responses to varying concentrations of 2-APB without or with 3 µM Dyc after full sensitization of TRPV3. (**E**) Concentration-response curves of 2-APB without or with Dyc. Data are shown as relative values to the current evoked by 300 µM 2-APB. Solid lines are fits to Hill equation, yielding EC$_{50}$ = 22.93 ± 0.02 µM and $n_H$ = 1.6 ± 0.1 without Dyc (n = 6); and EC$_{50}$ = 22.03 ± 0.86 µM and $n_H$ = 1.7 ± 0.1 with 3 µM Dyc (n = 6). (**F**) Dose-response curves normalized to its own maximum of each condition. (**G, H**) Representative whole-cell recordings for the sensitization of TRPV3 currents elicited by repeated applications of 100 µM 2-APB in the absence (**G**) and presence (**H**) of 5 µM Dyc. (**I**) Time courses toward the peak currents elicited by repeated application of 100 µM 2-APB with or without Dyc (n = 9). Currents were normalized to each initial values. (**J**) The 2-APB-evoked inward currents were reversibly inhibited by Dyc in primary mouse epidermal

*Figure 1 continued on next page*

*Figure 1 continued*

keratinocytes. Representative inward currents were firstly activated by repeated application of 300 µM 2-APB at the holding potential of –60 mV, and then inhibited by 5 or 30 µM Dyc or 10 µM ruthenium red (RR) as indicated. (**K**) Summary of relative currents elicited by 300 µM 2-APB with or without Dyc. (**L**) Dose dependence of Dyc effects on TRPV3 currents in cultured keratinocytes. The solid line corresponds to a fit by Hill equation with $IC_{50}$ = 5.2 ± 0.71 µM and $n_H$ = 2.4 ± 0.75 (n = 6). The dotted line indicates zero current level.

curves (*Figure 1F*). Therefore, dyclonine dose-dependently inhibits the response amplitudes of TRPV3 channel.

TRPV3 channel in physiological conditions has a low level of response to external stimuli, which is augmented during the sensitization process (i.e., repetitive stimulations, *Figure 1A*). In contrast, excessive upregulation of TRPV3 activity impairs hair growth and increases the incidence of dermatitis and pruritus in both humans and rodents. To determine whether dyclonine affects the process of TRPV3 sensitization, TRPV3-expressing cells were repeatedly exposed to 100 µM 2-APB without or with 5 µM dyclonine (*Figure 1G,H*). TRPV3 currents evoked by 2-APB alone took approximately eight repetitions to reach full sensitization level (*Figure 1I*). The presence of dyclonine significantly slowed down this process, requiring ~16 repetitions to reach the current level of full sensitization (*Figure 1H,I*). As expected, dyclonine also reduced the initial TRPV3 current (31.12 ± 2.86 pA/pF vs. 86.43 ± 5.9 pA/pF without dyclonine; p<0.001; n = 9 per condition).

As TRPV3 is highly expressed in keratinocytes, we further determined the inhibitory effect of dyclonine in primary mouse epidermal keratinocytes. After stabilizing the channel current by repeated application of 2-APB, we tested the inhibitory effect of 5 and 30 µM dyclonine (*Figure 1J*). On average, TRPV3 currents were reduced to 52% ± 7% and 13% ± 0.01% of control level by 5 and 30 µM dyclonine, respectively (*Figure 1K*), reaching the similar level of inhibition by the wide-spectrum TRP channel blocker ruthenium red (RR, *Figure 1J*). From the dose-response curve (*Figure 1L*), the $IC_{50}$ of dyclonine was assessed to be 5.2 ± 0.71 µM, with a Hill coefficient of $n_H$ = 2.4 ± 0.75 (n = 7). Thus, dyclonine effectively suppresses the activity of endogenous TRPV3 channels in mouse keratinocytes.

## Dyclonine is a potent inhibitor of TRPV3 channel

Next, we compared the inhibitory effect on TRPV3 of dyclonine to its impact on other TRP channels. TRPV1, TRPV2, TRPM8, and TRPA1 channels were expressed in HEK 293T cells and respectively activated by capsaicin, 2-APB, menthol, and allyl isothiocyanate (AITC). We observed that 10 µM dyclonine exhibited little inhibition on TRPV1, TRPV2, TRPM8, and TRPA1, but potently inhibited TRPV3 channel (*Figure 2A*). The corresponding reduction in current amplitude was 2% ± 1% for TRPV1, 6% ± 1% for TRPV2, 9% ± 2% for TRPM8, 5% ± 1% for TRPA1, compared with 87% ± 1% inhibition of TRPV3 current (*Figure 2B*). By applying a series of dyclonine concentrations, we derived dose-response curves (*Figure 2C*). The corresponding $IC_{50}$ values of dyclonine for inhibiting TRPV1, TRPV2, TRPM8, and TRPA1 channels (336.3 ± 12.0 µM, 36.5 ± 3.7 µM, 72.4 ± 10.9 µM, and 152.35 ± 16.3 µM, respectively) were one or two orders of magnitude higher than that for TRPV3 inhibition (3.2 ± 0.24 µM), indicating that dyclonine represents an effective inhibitor of TRPV3 channel.

The above results were obtained for mouse TRPV3. We further asked whether the inhibitory effect of dyclonine on TRPV3 is consistent across different species. Similarly, we performed whole-cell recordings in HEK 293T cells expressing hTRPV3 and frog TRPV3 (fTRPV3), respectively. They were activated to a stable level by repetitive 2-APB stimulation. Addition of dyclonine, indeed, efficiently suppressed the activation of both types of TRPV3 channel (*Figure 2D–I*). Dose-response curves for dyclonine inhibition yielded an $IC_{50}$ value of 16.2 ± 0.72 µM for hTRPV3 and 12.3 ± 1.6 µM for fTRPV3, respectively. Therefore, the inhibition of TRPV3 by dyclonine is conserved across species.

## Inhibition of TRPV3 by dyclonine is voltage-independent

To obtain a complete description of the inhibitory effect of dyclonine, we next investigated its voltage dependence using a stepwise protocol (*Figure 3A*). We measured membrane currents in TRPV3-expressing HEK 293T cells using a $Cs^+$-based pipette solution that blocks most outward $K^+$ channel current but permits measurement of outward conductance mediated by the nonselective TRPV3 channel. A low-concentration 2-APB (40 µM) activated small voltage-dependent currents with steady-state outward rectification, characteristic of TRPV3 currents in heterologous expression systems (*Figure 3A*). Addition of dyclonine in the extracellular solution significantly diminished TRPV3-mediated

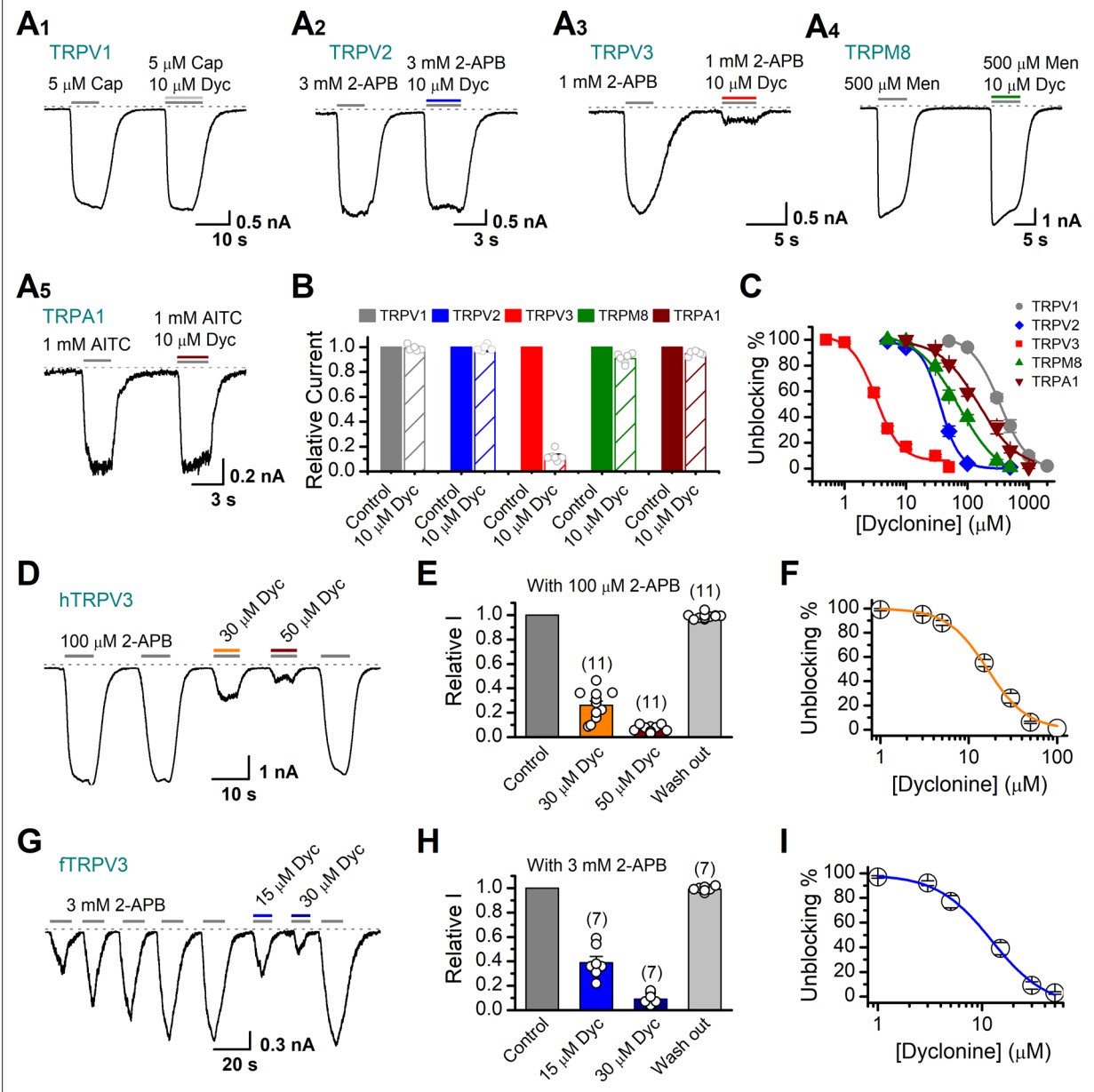

**Figure 2.** Dyclonine (Dyc) is a potent inhibitor of transient receptor potential vanilloid-3 (TRPV3) channel. (**A**) Representative inward current traces from whole-cell voltage-clamp recordings show the inhibitory effects of 10 µM Dyc on TRPV1 ($A_1$), TRPV2 ($A_2$), TRPV3 ($A_3$), TRPM8 ($A_4$), or TRPA1 ($A_5$) channels (Cap: capsaicin; Men: menthol). Bars represent duration of drug application. (**B**) Summary of relative currents before and after Dyc (10 µM) treatment. Numbers of cells are indicated in parentheses. (**C**) Dose-response curves of Dyc for inhibition of indicated ion channel currents. Solid lines represent fits by Hill equation, with $IC_{50}$ = 337.4 ± 19.4 µM and $n_H$ = 2.0 ± 0.31 for TPRV1 (n = 7), $IC_{50}$ = 31.1 ± 2.7 µM and $n_H$ = 2.9 ± 0.50 for TPRV2 (n = 8), $IC_{50}$ = 81.8 ± 12.7 µM and $n_H$ = 1.2 ± 0.20 for TRPM8 (n = 6), and $IC_{50}$ = 154.7 ± 15.6 µM and $n_H$ = 1.3 ± 0.15 for TRPA1 (n = 6). For comparison, the dose-response curve of TRPV3 channel from *Figure 1C* is displayed in red with $IC_{50}$ = 3.2 ± 0.24 µM and $n_H$ = 2.2 ± 0.32 (n = 6). (**D**) Suppression of 2-aminoethoxydiphenylborate (2-APB)-evoked currents by Dyc in a human TRPV3 (hTRPV3)-expressing HEK 293T cell. Representative inward current trace shows the reversible block effect of Dyc (30 and 50 µM) at the holding potential of –60 mV. (**E**) Summary of inhibition of hTRPV3 by Dyc. Membrane currents were normalized to the responses elicited by the saturated concentration of 2-APB (100 µM) alone. (**F**) Dose-response curve for Dyc on blocking of hTRPV3. Solid line represents a fit to a Hill equation, yielding $IC_{50}$ = 16.2 ± 0.72 µM and $n_H$ = 1.91 ± 0.14 (n = 11). (**G**) Inhibition of frog TRPV3 (fTRPV3) currents by Dyc. Representative whole-cell currents at –60 mV in a fTRPV3-expressing HEK 293T cell. After sensitization by repeated application of 3 mM 2-APB, the cell was exposed sequentially to 15 and 30 µM Dyc with 3 mM 2-APB. (**H**) Summary of inhibition of relative currents elicited by 3 mM 2-APB, 3 mM 2-APB with Dyc 15 or 30 µM. (**I**) Concentration-response curve of Dyc on the inhibition of fTRPV3 currents. Solid line represents a fit by a Hill equation, with $IC_{50}$ = 12.31 ± 1.6 µM and $n_H$ = 1.6 ± 0.34 (n = 7). The dotted line indicates zero current level.

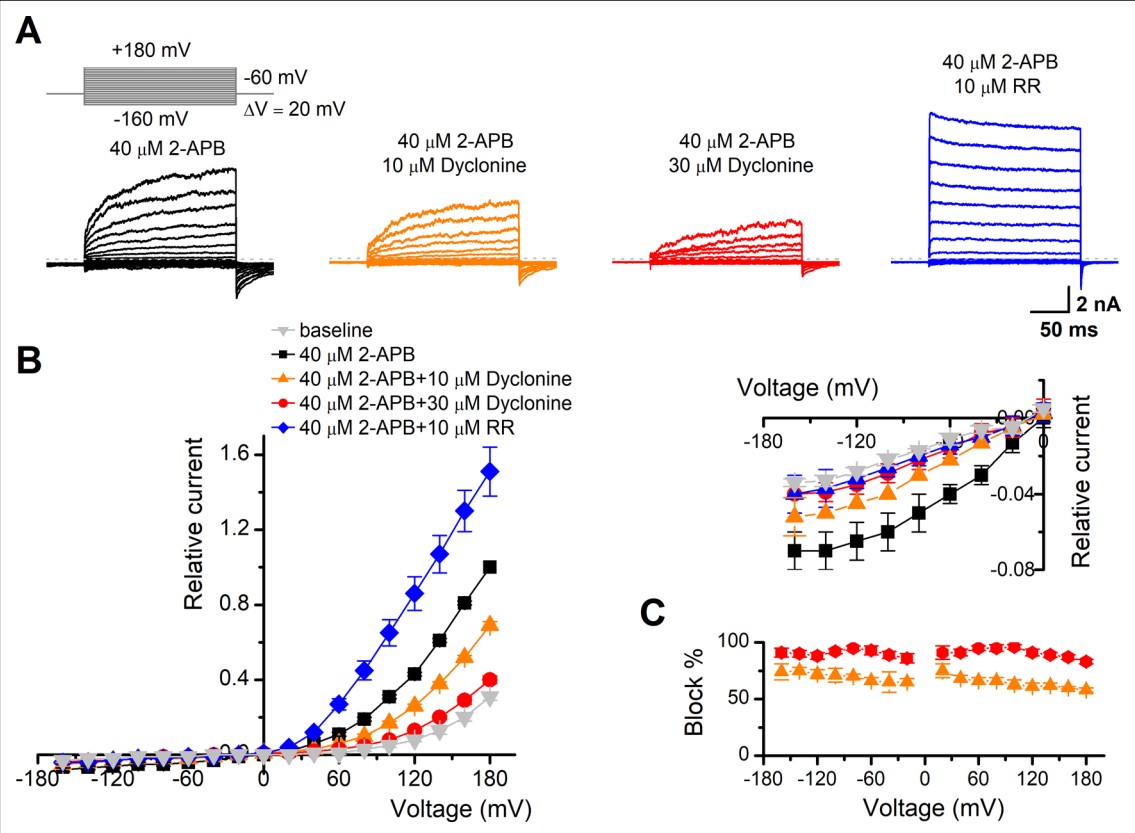

**Figure 3.** The inhibitory effect of dyclonine on transient receptor potential vanilloid-3 (TRPV3) channel is voltage-independent. (**A**) Representative whole-cell currents evoked by voltage steps (inset) together with 40 µM 2-aminoethoxydiphenylborate (2-APB) in the absence and presence of 10, 30 µM dyclonine or 10 µM ruthenium red (RR) in HEK 293T cells expressing mouse TRPV3. Currents were elicited with 200 ms test pulses ranging from –160 mV to +180 mV in 20 mV increments within the same cells, and the holding potential was –60 mV. Calcium-free standard bath solution and a CsCl-filled recording electrode were used. The dotted line indicates zero current level. (**B**) Current-voltage relations for data in (**A**). Current amplitudes were normalized to the maximum responses at +180 mV in the presence of 40 µM 2-APB. Each point represents mean values (± SEM) from eight independent cells. (Inset) The inhibition effects of dyclonine and RR on TRPV3 currents at negative holding potentials are magnified and displayed on the right. Note that dyclonine had an inhibitory effect on TRPV3 currents at both positive and negative potentials, but RR only inhibited TRPV3 channel currents at negative potentials while enhanced TRPV3 currents at positive potentials (blue trace). (**C**) Percentage block of TRPV3 currents by dyclonine (10 and 30 µM) as a function of membrane potential. Error bars represent SEM.

outward and inward currents (*Figure 3A*). By contrast, 10 µM RR, a broad TRP channel blocker, only inhibited TRPV3-mediated inward currents but enhanced outward currents (*Figure 3A*), which is consistent with early report (*Cheng et al., 2010*). Dyclonine inhibition of both inward and outward currents was further confirmed by the I-V curves derived from pooled data (*Figure 3B*). We found no significant difference inhibition at hyperpolarized voltages versus depolarized voltages, showing that the inhibition occurred independently of the membrane potential (*Figure 3C*). Together, relative to the wide-spectrum blocker RR, dyclonine more effectively inhibits TRPV3 channel in a voltage-independent manner.

## Inhibition of heat-activated TRPV3 currents by dyclonine

TRPV3 is a thermal-sensitive ion channel and has an activation threshold around 30–33°C (*Xu et al., 2002*). We therefore explored whether the heat-evoked TRPV3 currents can be also inhibited by dyclonine. We employed an ultrafast infrared laser system to control the local temperature near single cells; each temperature jump had a rise time of 1.5 ms and lasted for 100 ms. TRPV3 sensitization of the channel was induced by repeating a same temperature jump from room temperature to ~51°C (*Figure 4A*). TRPV3, expressed in HEK 293T cells, steadily responded to temperature jumps ranging from 30 to 51°C (*Figure 4B*). After pre-sensitization by repeated temperature jumps from room temperature to 52°C, application of dyclonine appreciably inhibited TRPV3 thermal currents

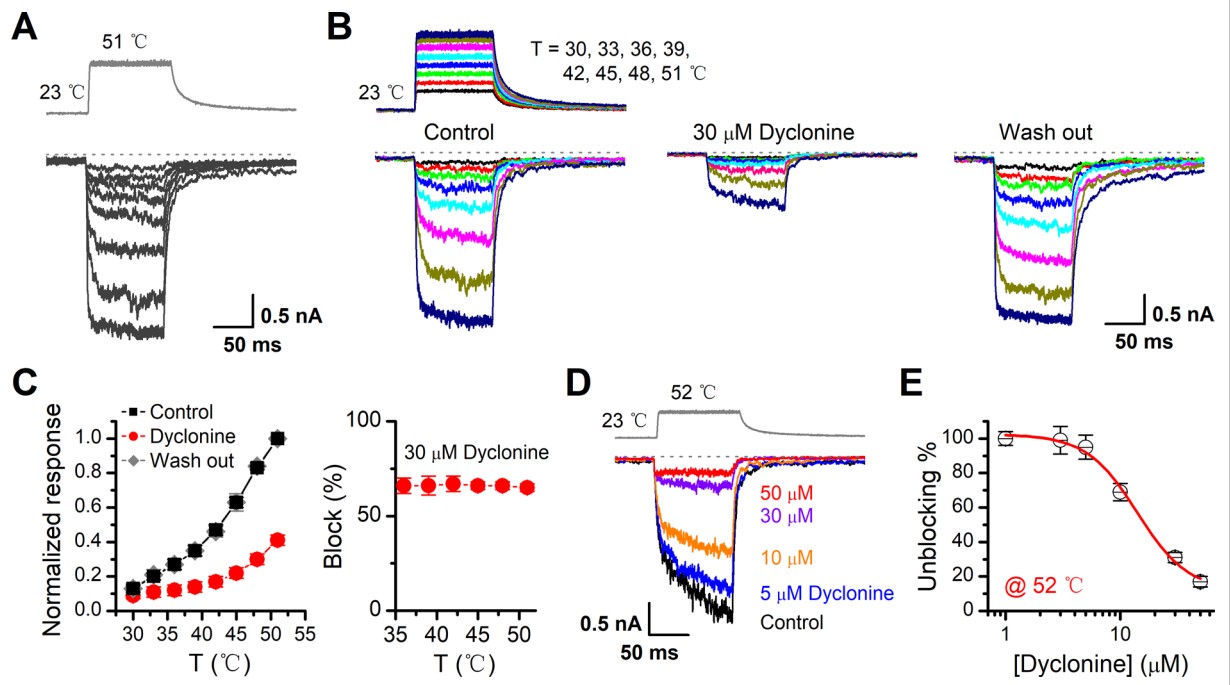

**Figure 4.** Inhibition of heat-activated transientreceptor potential vanilloid-3 (TRPV3) currents by dyclonine. (**A**) Sensitization of TRPV3 by heat. Heat-evoked TRPV3 currents in response to repeated temperature jumps. Temperature pulse generated by infrared laser diode irradiation was stepped from room temperature to 51°C in 1.5 ms and then clamped for 100 ms. (**B**) Effects of dyclonine on heat-activated TRPV3 currents. Heat-evoked current traces were recorded in whole-cell configuration, which were stabilized by sensitization of repeated fast temperature jumps as shown in (**A**). Temperature jumps shown on the top had a duration of 100 ms and a rise time of 1.5 ms. Bath solution with 0 or 30 μM dyclonine was applied by brief perfusion to the patch just before temperature stimulation on the same cells. (**C**) The average plot compares the temperature responses in the absence and presence of 30 μM dyclonine (left, n = 6). Currents were normalized by their maximum responses under control condition, respectively. Note that data from control and washout are superimposed. Percentage block of heat-evoked TRPV3 currents by 30 μM dyclonine as a function of temperature is shown on the right. (**D**) Representative inward currents evoked by a series of identical temperature jumps inhibited by dyclonine in a concentration-dependent manner. The temperature pulse (52°C) is shown in gray. Holding potential was –60 mV. (**E**) Dose dependence of dyclonine effects on heat-activated TRPV3 currents. The solid line represents a fit to Hill equation with $IC_{50} = 14.1 \pm 2.5$ μM and $n_H = 1.9 \pm 0.54$ (n = 10). All whole-cell recordings were got from TRPV3-expressing HEK 293T cells held at –60 mV.

(*Figure 4B,C*). The inhibitory effect of dyclonine was fully reversible as after its washing out the TRPV3 response recovered to the same level as control condition (*Figure 4C*). To determine the concentration dependence of dyclonine inhibition, TRPV3 currents were evoked by a same temperature jump from room temperature to ~52°C in the presence of 1, 3, 5, 10, 30, and 50 μM dyclonine (*Figure 4D*). The $IC_{50}$ of dyclonine on TRPV3 inhibition was assessed to be $14.02 \pm 2.5$ μM with a Hill coefficient of $n_H = 1.9 \pm 0.54$, according to the dose-response curve fitting (*Figure 4E*). These results thus indicate that dyclonine dose-dependently suppresses heat-evoked TRPV3 currents.

## Dyclonine inhibited hyperactive TRPV3 mutants and rescued cell death

It has previously been reported that gain-of-function mutations, G573S and G573C, of TRPV3 are constitutively active and their expression causes cell death (*Xiao et al., 2008*). We first examined the effect of dyclonine on the electrophysiological activity of mutants. We transfected the inducible cDNA constructs encoding respectively the GFP-tagged wild-type (WT) TRPV3, G573S, or G573 mutant into T-Rex 293 cells and then applied 20 ng/ml doxycycline to induce the gene expression. As illustrated in *Figure 5A, B*, whole-cell recordings from G573S or G573C expressed in T-Rex 293 cells show that the spontaneous currents noticeably appeared when changing the holding potential from 0 mV to –60 mV, and application of 2-APB further increased the channel currents. In each patch, 20 μM RR was applied extracellularly to obtain remaining leak currents. By subtracting leak currents, we found that spontaneous activities from G573S and G573C were reduced by 74% ± 3% (n = 6) and 71% ± 2% (n = 6) by 10 μM dyclonine, respectively (*Figure 5C*). Also, the presence of dyclonine significantly inhibited

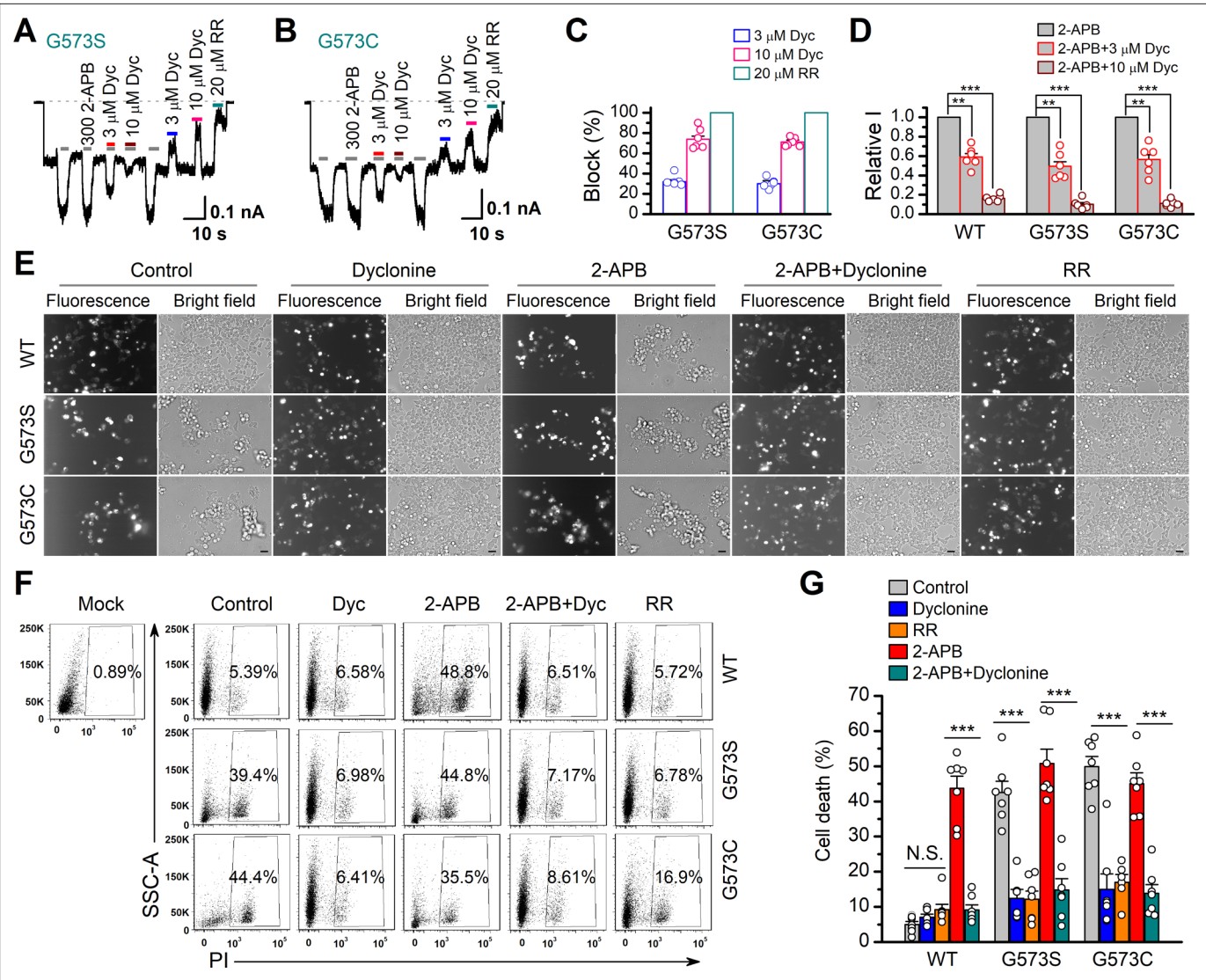

**Figure 5.** Dyclonine (Dyc) rescued cell death caused by expression of overactive transient receptor potential vanilloid-3 (TRPV3) mutant. (**A, B**) Effects of Dyc on whole-cell currents recorded from TRPV3 (G573S) and TRPV3 (G573C) expressed in T-Rex 293 cells, showing that Dyc (3 and 10 μM) reversibly inhibited the response to 300 μM 2-aminoethoxydiphenylborate (2-APB) and the spontaneous activities at –60 mV. 20 μM ruthenium red (RR) was applied for subtracting leak currents. Bars represent duration of drug application. (**C**) Averaged inhibition of spontaneous activities of G573 mutants by Dyc and RR. (**D**) Summary of relative whole-cell currents of TRPV3 (wild-type [WT]), G573S, and G573S with or without Dyc treatment. Error bars represent SEM. (**E**) Bright-field and fluorescence images showing the cell survival. The GFP-tagged TRPV3 (WT) and two mutants (G573C and G573S) in pcDNA4/TO vector were respectively transfected into T-Rex 293 cells, and then treated with doxycycline (20 ng/ml) for 16 hr post-transfection to induce gene expression in the presence of drugs as indicated. Images of cells were taken at 12 hr after induction. Scale bar, 50 μm. (**F**) Flow cytometry analysis of the percentage of dead cells. Cells transfected with the desired plasmids are as indicated. After the gene expression induced by doxycycline, the cells were treated with Dyc (50 μM), 2-APB (30 μM), RR (10 μM), or the combination of 2-APB and Dyc, and then stained with propidium iodide, followed by flow cytometry to analyze cell survival. (**G**) Summary plots of cell death rates under different treatments. Data were averaged from seven independent experiments. *** p<0.0001.

300 μM 2-APB-evoked responses to 10% ± 2% (G573S, n = 6) and 11% ± 1% (G573C, n = 6) of control level (*Figure 5D*), respectively. As both mutant TRPV3 channels are effectively inhibited by dyclonine, we next explored whether it can rescue the cell death caused by these gain-of-function mutants. Cells expressing G573S or G573S were exposed to different pharmacological drugs (dyclonine, 2-APB, 2-APB and dyclonine, or RR). Cell death was recognized by the narrow and contracted footprints in bright-field images, and the protein expression meanwhile monitored by GFP fluorescence. As shown in *Figure 5E*, massive cell death was seen in cells that expressed G573C and G573S TRPV3 mutants

but not those expressing the wild-type TRPV3. Addition of dyclonine largely prevented the cell death while not causing change in the expression of TRPV3 channels (*Figure 5E*), indicating that dyclonine decreased the cytotoxicity caused by the gain-of-function mutants. We further performed flow cytometry analysis and observed that the cell death ratio was maintained at low level (4.96% ± 0.87%, n = 7) in cells expressing WT TRPV3 (*Figure 5F*). By contrast, the expression of G573S or G573C mutant significantly increased the cell death ratio to 45.36% ± 5.79% (n = 7) and 52.74% ± 4.94% (n = 7), which were effectively reduced by dyclonine (50 μM) to 12.45% ± 2.54% (n = 7) and 14.98% ± 4.40% (n = 7), respectively. The cell-protective effect of dyclonine was mirrored by the general TRP channel blocker RR (*Figure 5E–G*). As expected, activation of TRPV3 channels with the agonist 2-APB caused significant cell death even in cells expressing WT channel and exacerbated the cell death in those expressing the mutant channel G573S or G573C (*Figure 5G*). Application of dyclonine also reversed the cell death caused by 2-APB activation (9.12% ± 1.42% vs. 43.73% ± 3.46% for WT condition, 17.68% ± 5.66% vs. 53.60% ± 5.88% for G573S, and 13.85% ± 2.49% vs. 47.91% ± 5.54% for G573C after and before addition of dyclonine). Collectively, these results indicate that dyclonine rescues cell death by inhibiting the excessive activity of TRPV3 channel.

## Dyclonine targets TRPV3 in vivo and ameliorates scratching behavior

TRPV3 is highly expressed in skin keratinocytes, whose hyperactivity causes pruritic dermatitis and scratching behavior. We next examined in vivo the therapeutic effect of dyclonine on TRPV3 hyperactivity-caused scratching behavior in mouse model. Itching-scratching behavior was induced by pharmacological activation of TRPV3 channel by a natural compound carvacrol derived from oregano (*Cui et al., 2018*). The number of scratching bouts was quantified every 5 min (*Figure 6A*), and also summed over a 30 min observation period (*Figure 6B*). Intradermal injection of carvacrol (0.1%, 50 μl) in WT TRPV3 mice caused significant increases in the accumulated scratching bouts (137.2 ± 33.9) compared to the control group receiving normal saline (0.9% NaCl, 3.8 ± 1, n = 6, p<0.001; *Figure 6B*). By contrast, intradermal injection of carvacrol (0.1%, 50 μl) did not elicit a remarkable change in the number of scratching bouts in TRPV3$^{-/-}$ mice (*Figure 6A,B*), supporting that carvacrol caused itching-scratching behavior via TRPV3 activation (*Cui et al., 2018*). To investigate whether dyclonine could alleviate carvacrol-evoked acute itch, we made an intradermal injection of dyclonine into the mouse neck 30 min before the injection of carvacrol into the same site. As illustrated in *Figure 6C,D*, administration of 50 μl dyclonine at 1, 10, and 50 μM concentrations appreciably reduced the scratching bouts to 130.0 ± 20.3, 82.0 ± 15.0, and 18.0 ± 8.0 from 137.8 ± 18.3 (n = 6), respectively. We also carried out whole-cell recordings in TRPV3-expressing HEK 293T cells to further confirm the inhibitory effect of dyclonine on TRPV3 currents activated by carvacrol. Similar to that observed with the inhibition of 2-APB-evoked TRPV3 currents (*Figure 1A–C*), dyclonine also inhibited carvacrol-activated TRPV3 currents in a concentration-dependent manner with IC$_{50}$ = 3.5 ± 0.34 μM following sensitization by repeated application of 300 μM 2-APB (n = 8, *Figure 6E,F*), implying that the itching caused by carvacrol is mainly due to the activation of TRPV3. Hence, dyclonine ameliorates TRPV3 hyperactivity-caused scratching in a concentration-dependent manner. In contrast, dyclonine (10 μM) showed little effect on electrophysiological responses in mouse dorsal root ganglia (DRG) and trigeminal ganglia (TG) neurons (*Figure 6—figure supplement 1*). This observation is in line with the absence of TRPV3 in mouse DRGs (*Peier et al., 2002*) and suggests that the invio effect of dyclonine arises from the targeting of keratinocyte TRPV3 channels.

We also used WT and TRPV3 KO mice to examine the effect of dyclonine on thermal nociceptive responses to the noxious temperature 55°C. In WT mice, dyclonine exhibited a tendency to reduce the nociceptive response (*Figure 6—figure supplement 1*). TRPV3 KO reduced mice nociceptive response to heating compared to WT mice (55°C; comparison between gray bars in *Figure 6—figure supplement 1E*). However, in TRPV3 KO mice, dyclonine showed no further effect, showing that dyclonine mainly targets TRPV3 in vivo. These observations also suggest that TRPV3 partially contributes to pain sensation in thermal nociception, in consistency with the temperature-dependent responses of TRPV3 channel (*Figure 4*).

## Effects of dyclonine on single TRPV3 channel activity

We then examined the functional and molecular mechanisms underlying the inhibition of TRPV3 by dyclonine. To distinguish whether such inhibition arises from the changes in channel gating or

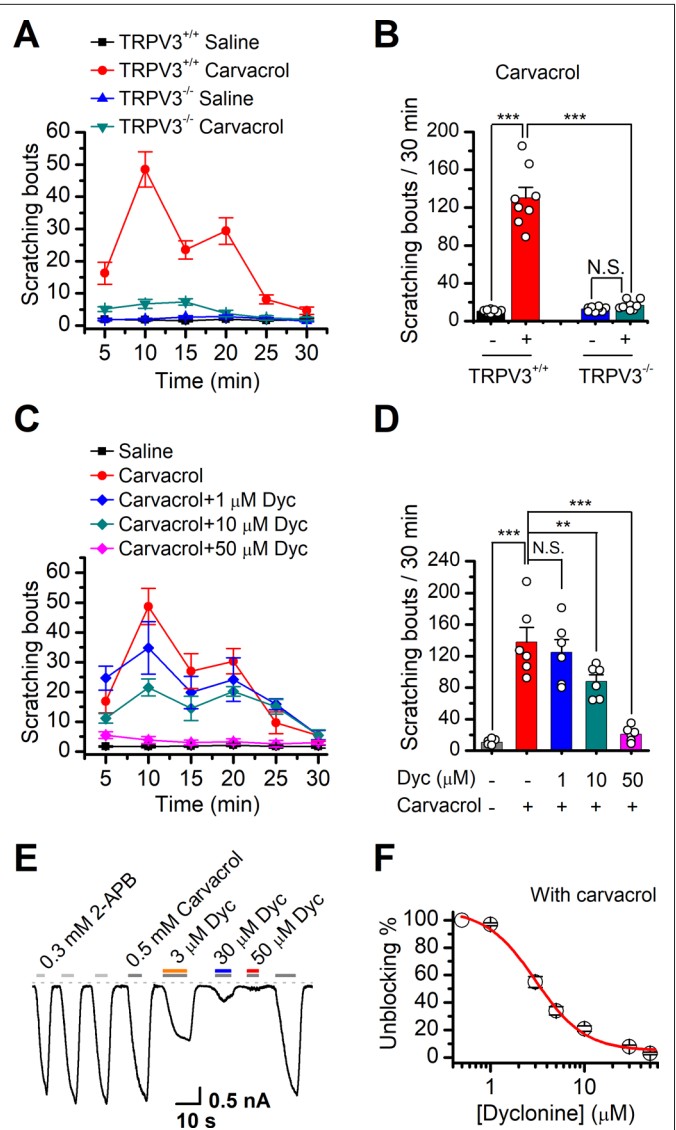

**Figure 6.** Dyclonine (Dyc) suppresses scratching behavior induced by carvacrol. (**A**) Summary of the time courses of neck-scratching behaviors in wild-type transientreceptor potential vanilloid-3 (TRPV3) and TRPV3 knock out (C57BL/6) mice after intradermal injection of 50 µl carvacrol (0.1%) or normal saline (0.9% NaCl) containing 0.1% ethanol into the mouse neck. Time for scratching bouts was plotted for each 5 min interval over the 30 min observation period. (**B**) Quantification of the cumulative scratching bouts over 30 min under different treatments, showing that intradermal injection of carvacrol elicited a remarkable increase in the number of scratching bouts in TRPV3$^{+/+}$ but not TRPV3$^{-/-}$ mice (n = 6; N.S.: no significance; *p<0.05; **p<0.01; ***p<0.001, by one-way ANOVA). (**C**) Time courses of neck-scratching behaviors in response to intradermal injection of 50 µl carvacrol (0.1%), with pretreatment of normal saline (0.9% NaCl), or different concentrations (1, 10, and 50 µM) of Dyc in the same site. (**D**) Summary plots of the number of scratching bouts over 30 min under different treatments as indicated, showing that Dyc dose-dependently alleviated carvacrol-evoked acute itch (n = 6; N.S.: no significance; *p<0.05; **p<0.01; ***p<0.001, by one-way ANOVA). (**E**) Inhibition of carvacrol-evoked currents by Dyc in a representative HEK 293T cell expressing TRPV3. After sensitization by repeated application of 300 µM 2-aminoethoxydiphenylborate (2-APB), the cell was exposed to 3, 30, or 50 µM Dyc with 500 µM carvacrol as indicated by the bars. Membrane currents were recorded in a whole-cell configuration, and the holding potential was –60 mV. (**F**) The dose-response curve for Dyc inhibition of carvacrol-evoked TRPV3 currents was fitted by Hill equation (IC$_{50}$ = 3.5 ± 0.34 µM and n$_H$ = 2.1 ± 0.41, n = 8).

The online version of this article includes the following figure supplement(s) for figure 6:

**Figure supplement 1.** Effects of dyclonine on the excitability of mouse dorsal root ganglia (DRG) and trigeminal ganglia (TG) neurons as well as the transient receptor potential vanilloid-3 (TRPV3)-mediated nociceptive behavior.

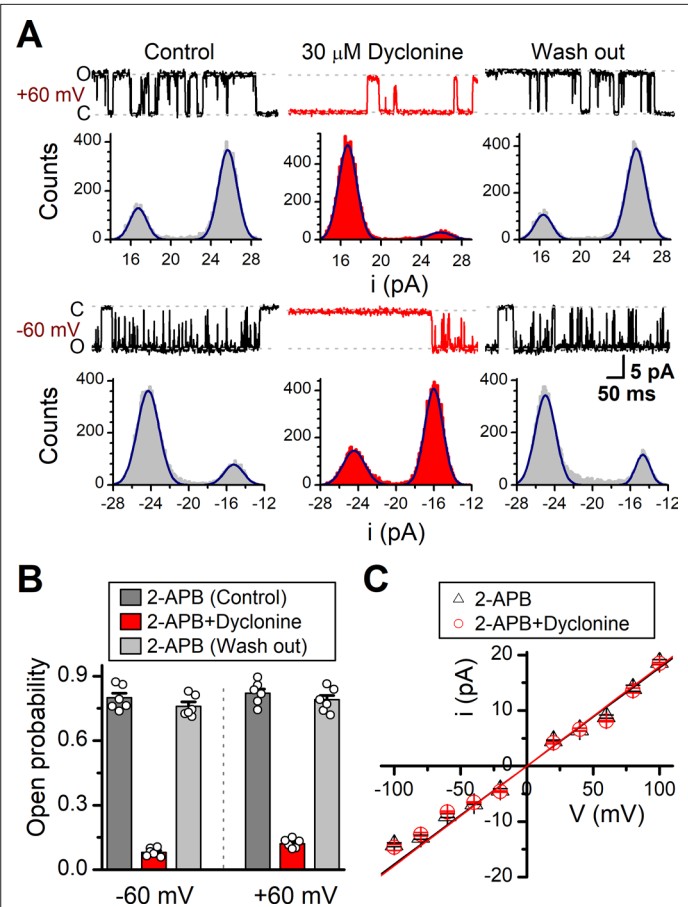

**Figure 7.** Effects of dyclonine on single-channel properties of transient receptor potential vanilloid-3 (TRPV3). (**A**) Single-channel currents of TRPV3 were recorded from inside-out membrane patches of HEK 293T cells at two membrane potentials (± 60 mV) in symmetrical 150 mM NaCl and were low-pass filtered at 2 kHz. Currents were evoked by 10 μM 2-aminoethoxydiphenylborate (2-APB) in the absence and presence of dyclonine (30 μM) after sensitization induced by repetitive 300 μM 2-APB. All-point amplitude histograms of single-channel currents were shown below the current traces. The histograms were fit to sums of two Gaussian functions to determine the average amplitudes of currents and the open probabilities. Dotted lines indicate the opened channel state (**O**) and the closed channel state (**C**), respectively. (**B**) Summary of effects of dyclonine on TRPV3 single-channel open probability. Dyclonine (30 μM) decreased TRPV3 open probability from 0.8 ± 0.02 to 0.08 ± 0.01 at –60 mV (n = 6), and from 0.82 ± 0.02 to 0.12 ± 0.01 at +60 mV (n = 6), respectively. (**C**) I-V relationships of TRPV3 single-channel current evoked by 10 μM 2-APB without (black triangles) and with 30 μM dyclonine (red circles). Unitary conductance measured by fitting a linear function were 163.6 ± 6.4 pS and 179.2 ± 5.5 pS for before and after treatment by dyclonine, respectively.

conductance, we measured single-channel activity. Single-channel recordings were performed in an inside-out patch that was derived from HEK 293T cells expressing the mouse TRPV3 (*Figure 7*). Currents were evoked by 10 μM 2-APB in the absence and presence of dyclonine (30 μM) after sensitization induced by 300 μM 2-APB at a holding potential of either +60 mV or –60 mV (*Figure 7A*). To quantify the changes, we constructed all-point histograms and measured the open probabilities and the unitary current amplitudes by Gaussian fitting. We observed that the single-channel open probability was largely decreased by dyclonine from 0.8 ± 0.02 to 0.08 ± 0.01 at –60 mV and from 0.82 ± 0.02 to 0.12 ± 0.01 at +60 mV (n = 6), respectively (*Figure 7B*). Statistical analysis, however, revealed that dyclonine had no effect on single TRPV3 channel conductance (163.6 ± 6.4 pS vs. 179.2 ± 5.5 pS for before and after dyclonine treatment; *Figure 7C*).

## The mechanism underlying the inhibition of TRPV3 by dyclonine

In order to understand the molecular mechanism underlying the blockade of TRPV3 by dyclonine, we utilized in silico docking to predict their interactions. The inhibitory effect of drugs on ion channels is usually achieved in three ways, competitively binding with agonists, negative allosteric regulation, or directly blocking the channel pore. Dyclonine inhibited TRPV3 currents evoked by both 2-APB (*Figure 1*) and heat (*Figure 4*), implying that dyclonine is not a competitive antagonist. In addition, the voltage independence of dyclonine inhibition and the fact that dyclonine is a positive charged alkaloid suggest that dyclonine is not simply an open channel blocker. Previous studies have demonstrated that local anesthetics inhibit voltage-gated sodium channels through a common drug-binding region within the channel pore (*Tikhonov and Zhorov, 2017*). We therefore suspected that the inhibition effect of dyclonine is also due to its allosteric interaction with specific residues within the aqueous pore of TRPV3. The grid file of in silico docking was then constructed to examine residues in the upper pore region and the central cavity of TRPV3 (*Figure 8—figure supplement 1A*); the best receptor–ligand complex was evaluated using the extra precision (XP) scoring. Ligand clusters derived from XP docking suggested three potential TRPV3/dyclonine binding modes (BMs): $BM_A$, $BM_B$, and $BM_C$ (*Figure 8A,B*). Moreover, residues within 10 Å of dyclonine poses were extensively refined using induce-fit-docking (IFD) based on mTRPV cryo-EM structure (*Singh et al., 2018*; *Figure 8A*, *Figure 8—figure supplement 1B*). $BM_B$ and $BM_C$ modes predicted that dyclonine occupies the ion permeation pathway behaving as an open channel blocker. This, however, contradicts the fact that dyclonine is a positive charged alkaloid (*Figure 8B*) and its inhibition effect is voltage-independent (*Figure 3*). Hence, $BM_B$ and $BM_C$ binding modes appear unlikely. Nevertheless, mutants in key residues in these two binding sites diversely affected the inhibition of dyclonine (I637A, $IC_{50}$ = 6.1 ± 0.43 µM; F666A, $IC_{50}$ = 414.5 ± 15.7 µM; I674A, $IC_{50}$ = 15.1 ± 2.1 µM, *Figure 8—figure supplement 1E–H*), suggesting that the pore region is crucial for dyclonine inhibition.

$BM_A$ mode shows that dyclonine makes contacts with the cavity formed by the pore loop and S6-helix of TRPV3 (*Figure 8A,B*). Structures assigned to *apo* and open states revealed remarkable allosteric changes and cavity size reduction in these regions (*Figure 8—figure supplement 1G, H*), supporting the rationality of the $BM_A$ mode.

To further delineate dyclonine-interacting residues, we systematically mutated the residues in the cavity of TRPV3 channel predicted by $BM_A$ binding mode. Among the mutants, mutations L630W, N643A, I644W, and L655A greatly reduced the inhibitory effect of dyclonine, whereas the mutants L642A and I659A showed higher sensitivity to dyclonine than WT channel (*Figure 8C,D*). The dose-response curves were fitted with a Hill equation, and the corresponding $IC_{50}$ values for each TRPV3 mutant were as follows: $IC_{50}$ = 286.7 ± 10.4 µM for L655A; $IC_{50}$ = 30.8 ± 2.2 µM for L630W; $IC_{50}$ = 37.7 ± 5.1 µM for N643A; $IC_{50}$ = 26.1 ± 2.8 µM for I644W; $IC_{50}$ = 0.25 ± 0.02 µM for L642A; and $IC_{50}$ = 0.56 ± 0.06 µM for I659A compared to $IC_{50}$ = 3.2 ± 0.24 µM for WT TRPV3 (*Figure 8D,E*). Notably, all mutant channels except L639A were functional and produced robust responses to 2-APB (*Figure 8F*). Covalent modification of L630C, F633C, and L642C, with side chains toward the proposed binding site, using MTSET (2-(trimethylammonium) ethyl methanethiosulfonate, bromide), an MTS reagent with bulk positive side chain, significantly decreased 2-APB-idnuced current in the mutated mTRPV3 channels (*Figure 8G,H*). The reduction reagent dithiothreitol (DTT) rescued this inhibitory effect, indicating that the interruption of the allostery of the pore cavity has impaired the channel activation of mTRPV3 (*Figure 8G,H*). In contrast, MTSET treatment had no effect on the activation of WT TRPV3 (*Figure 8H*). Along the same line, MTSET modification caused reduced dyclonine blockade in L630C, F633C, and L642C but not WT TRPV3, and DTT restored the blockage of dyclonine in these mutants (*Figure 8I*), implying that dyclonine-mediated inhibition is mediated by the region predicted by $BM_A$ binding mode. Together, our results suggest that dyclonine interacts with the pore cavity of TRPV3 likely behaving as a negative allosteric modulator.

## Discussion

As a multimodal sensory channel, TRPV3 is abundantly expressed in keratinocytes and implicated in inflammatory skin disorders, itch, hair morphogenesis, and pain sensation (*Broad et al., 2016*). Human Olmsted syndrome has been linked to the gain-of-function mutations of TRPV3 (*Agarwala et al., 2015*; *Lai-Cheong et al., 2012*; *Lin et al., 2012*). Synthetic and natural compounds, like isopentenyl

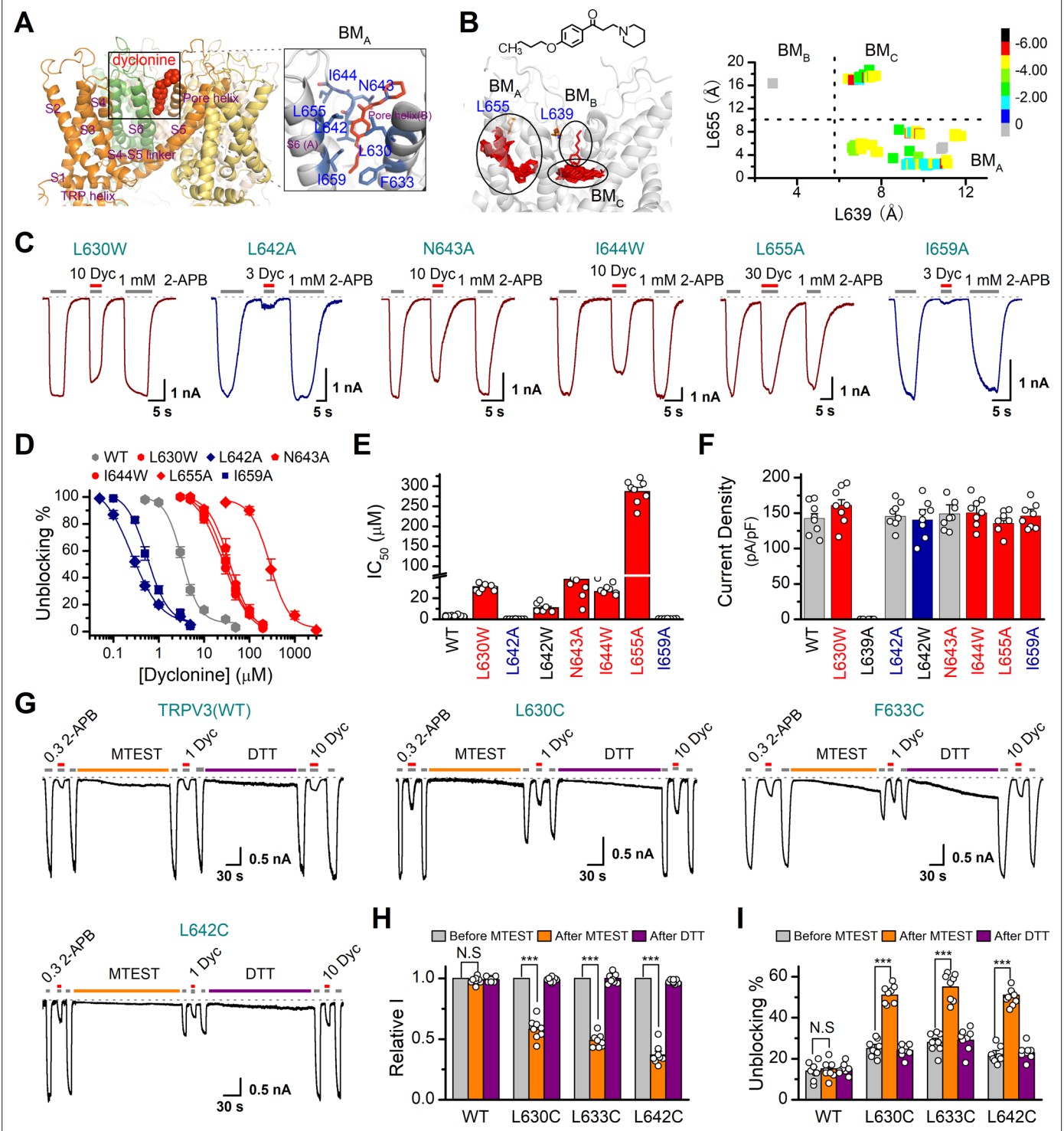

**Figure 8.** Molecular residues involved in dyclonine-transient receptor potential vanilloid-3 (dyclonine-TRPV3) interaction. (**A**) Overall view of the mTRPV3-dyclonine complex. Three putative binding modes (BMs) for dyclonine in the pore cavity of mTRPV3 channel (PDB ID code: 6DVZ) are denoted as BM$_A$, BM$_B$, and BM$_C$ (please find the details in the text), with the expanded view of BM$_A$ shown on the right. Four subunits of the tetramer are distinguished by different colors, and dyclonine in a schematic structure is shown in red. (**B**) (Left) Potential docking poses of dyclonine and TRPV3 channel. (Right) Cluster analysis showing all BMs distributed into three clusters, BM$_A$, BM$_B$, and BM$_C$. (**C**) Representative whole-cell recordings show reversible blocking of 2-aminoethoxydiphenylborate (2-APB) (1 mM)-evoked responses by dyclonine (3, 10, or 30 μM) in HEK 293T cells expressing mutant TRPV3 channels as indicated, respectively. The combination of 3, 10, or 30 μM dyclonine and 2-APB was applied following the control currents evoked by a saturated concentration of 2-APB (1 mM, initial gray bar). Holding potential was –60 mV. Bars represent duration of stimuli. (**D**)

*Figure 8 continued on next page*

*Figure 8 continued*

Concentration-response curves of dyclonine on inhibition of the TRPV3 mutants. Solid lines represent fits by a Hill equation, with the half-maximal inhibitory concentration (IC$_{50}$) shown in (**E**). For comparison, the dose-response curve of wile-type channel is displayed in gray. Four point mutations (L630W, N643A, I644W, and L655A) reduced the inhibitory efficiency of dyclonine, while the other two point mutations (L642A and I659A) enhanced the inhibitory effects of dyclonine on TRPV3 currents. (**F**) Average current responses of mutant channels compared with wild-type TRPV3 channels. Each substitution of putative residues except L639A retained their normal responses to 2-APB. Numbers of cells are indicated in parentheses. (**G**) Modulation of thiol-oxidizing and disulfide-reducing agents on the inhibitory effects of dyclonine. Whole-cell recordings from the wild-type TRPV3 and the mutants expressed in HEK 293 T cells, showing the effects of (2-(trimethylammonium) ethyl methanethiosulfonate, bromide) (MTSET) and dithiothreitol (DTT) on the responses to 2-APB with or without dyclonine after sensitization induced by 300 µM 2-APB. MTSET (1 mM) and DTT (10 mM) were locally applied for ~3 min to probe the accessibility, respectively. The responsiveness to 2-APB or 2-APB plus dyclonine was subsequently examined before and after treatments. Holding potential was –60 mV. (**H**) Summary of inhibition of relative currents elicited by 300 µM 2-APB, 300 µM 2-APB with dyclonine 10 or 1 µM. (**I**) Summary of inhibitory effects of dyclonine before and after treatments with MTSET and DTT. The dotted line indicates zero current level in all cases. Error bar represents SEM. N.S.: no significance; *p<0.05; **p<0.01; ***p<0.001.

The online version of this article includes the following figure supplement(s) for figure 8:

**Figure supplement 1.** Residues in the transient receptor potential vanilloid-3 (TRPV3) channel pore for interacting with dyclonine.

**Figure supplement 2.** Alignment of the pore-region sequences.

pyrophosphate (*Bang et al., 2011*), 17(R)-resolvin D1 (*Bang et al., 2012*), forsythoside B (*Zhang et al., 2019*), diphenyltetrahydrofuran osthole (*Higashikawa et al., 2015*), and RR (*Xu et al., 2002*), have been proposed to inhibit TRPV3 channels. Due to either or both the lack of targeting specificity and the clinical application, their remedial potential remains to be determined. Hence, identifying and understanding clinical pharmaceutics that target TRPV3 channels will help to conceive therapeutic interventions.

Dyclonine is a topical antipruritic agent and has been used for clinical treatment of itching and pain for decades (*Gargiulo et al., 1992*; *Greifenstein et al., 1956*). While the therapeutic effect of dyclonine has been attributed to the inhibition of cell depolarization, the underlying mechanisms have not been fully understood. In the present study, we provide several tiers of evidence that dyclonine potently inhibits TRPV3 channel. Such inhibition was observed for TRPV3 responses to both chemical and thermal activation, suggesting that dyclonine is a condition-across inhibitor. Accordingly, dyclonine efficiently blocked the excessive activation of TRPV3 mutants and prevented cell death. Single-channel recordings revealed that dyclonine effectively suppresses the channel open probability without changing single-channel conductance. These data not only supplement a molecular mechanism for the therapeutic effect of dyclonine, but also suggest its application to curb TRPV3-related disorders. Using mouse model, we indeed observed that dyclonine ameliorates the TRPV3 hyperactivity-caused itch/scratching behaviors, indicating its therapeutic inhibition effect being maintained in vivo. As TRPV3 responds to moderate temperatures (30–40°C), dyclonine may thus be used to alleviate skin inflammations persisted in physiological temperatures. Also, as a clinical drug dyclonine has been widely used and thus has shown its safety to human body (*Gargiulo et al., 1992*; *Sahdeo et al., 2014*). In addition, as a potent inhibitor, dyclonine can also been a research tool to dissect the physiological and pathological characteristics of TRPV3 channel. While dyclonine effectively inhibits TRPV3 channels, our current results do not exclude its targeting of other molecular pathways. For instance, voltage-gated sodium channels have been shown to be inhibited by local anesthetics including dyclonine (*Sahdeo et al., 2014*; *Tikhonov and Zhorov, 2017*).

The current data also provide clues on the molecular mechanism underlying the inhibition of TRPV3 by dyclonine. The residues within the pore loop and S6-helix of TRPV3, as suggested by BM$_A$ binding mode, create a functional 'hotspot' contributing to the inhibition of dyclonine. Chemical modification experiment further confirmed the importance of this 'hotspot' to channel gating and dyclonine inhibition. Interestingly, the size of pocket BM$_A$ is distinct in the apo/resting and open states. Likely, binding of dyclonine into this pocket could prevent the structural rearrangements of pore loop during TRPV3 gating, implying that dyclonine behaves as a negative allosteric modulator. Although similar pockets can also be observed on other TRP channels, the amino acids that make up the pocket and the precise shape of the pocket are diverse (*Figure 8—figure supplement 2*). This may be the reason why TRPV3 is targeted by dyclonine (*Liao et al., 2013*; *Shimada et al., 2020*; *Singh et al., 2018*). F666 is located below the upper filter and behaves with a bulky hydrophobic side chain, which may play a role in maintaining the shape of BM$_A$ at the open state. This may be the reason why F666A is capable of

decreasing the inhibition of dyclonine. Our current study revealed critical residues located within the pore cavity of TRPV3 that regulate dyclonine inhibition, yet the possibility exists that dyclonine inhibition is mediated by indirect mechanisms involving interactions with other residues. Nevertheless, the molecular sites uncovered by the present study would be instrumental in pinpointing the dyclonine-TRPV3 interaction at the molecular level, thereby developing specific therapeutics for chronic pruritus, dermatitis, and skin inflammations.

# Materials and methods

Key resources table

| Reagent type (species) or resource | Designation | Source or reference | Identifiers | Additional information |
|---|---|---|---|---|
| Species (*Mus musculus*) | *Trpv3*$^{-/-}$ mice | **Wang et al., 2021** | PMID:32535744 | C57BL/6J background |
| Cell line (*Homo sapiens*) | HEK 293T | ATCC | Cat.#:CRL-3216 | |
| Cell line (*Homo sapiens*) | T-Rex 293 | Thermo Fisher | Cat.#:R71007 | |
| Chemical compound | 2-APB | Sigma-Aldrich | Cat.#:D9754 | TRPV1-3 agonist |
| Chemical compound | Carvacrol | MedChemExpress | Cat.#:499752 | TRPV3 agonist |
| Chemical compound | Menthol | Sigma-Aldrich | Cat.#:M278 | TRPM8 agonist |
| Chemical compound | Capsaicin | MedChemExpress | Cat.#: HY10448 | TRPV1 agonist |
| Chemical compound | AITC | Sigma-Aldrich | Cat.#:377430 | TRPA1 agonist |
| Chemical compound | Ruthenium red | Sigma-Aldrich | Cat.#:R2751 | TRP channel inhibitor |
| Chemical compound | Poly-L-lysinehydrochloride | Sigma-Aldrich | Cat.#:2658 | |
| Chemical compound | MTEST | MedChemExpress | Cat.#:690632554 | |
| Chemical compound | DTT | Sigma-Aldrich | Cat.#:3483123 | |
| Chemical compound | Dyclonine | MedChemExpress | Cat.#:536436 | |
| Software, algorithm | Patchmaster | HEKA Electronics | | |
| Software, algorithm | OriginPro | Originlab.com | | |
| Software, algorithm | Clampfit 10 | Molecular Devices | | |
| Software, algorithm | SigmaPlot 10 | SPSS Science | | |

## cDNA constructs and transfection in HEK 293T cells

The WT mouse TRPV3 (mTRPV3), human TRPV3 (hTRPV3), rat TRPV1, rat TRPV2, rat TRPM8, and mouse TRPA1 cDNAs were generously provided by Dr. Feng Qin (State University of New York at Buffalo, Buffalo, USA). The GFP-mTRPV3 WT and the mutants (mTRPV3-G573S and mTRPV3-G573C) in pcDNA4/TO vector were gifts from Dr. Michael X Zhu (The University of Texas Health Science Center at Houston, Houston, USA). The WT fTRPV3 was kindly provided by Dr. Makoto Tominaga (Department of Physiological Sciences, SOKENDAI, Okazaki, Japan). All mutations were made using the overlap-extension polymerase chain reaction method as previously described (*Tian et al., 2019*). The resulting mutations were then verified by DNA sequencing. HEK 293T and T-Rex 293 cells were grown in Dulbecco's modified Eagle's medium (DMEM, Thermo Fisher Scientific, MA) containing 4.5 mg/ml glucose, 10% heat-inactivated fetal bovine serum (FBS), 50 units/ml penicillin, and 50 mg/ml streptomycin, and were incubated at 37°C in a humidified incubator gassed with 5% $CO_2$. For T-Rex 293, blasticidin S (10 μg/ml) was also included. Cells grown into ~80% confluence were transfected with the desired DNA constructs using either the standard calcium phosphate precipitation method or lipofectamine 2000 (Invitrogen, Carlsbad, CA) following the protocol provided by the manufacturer. Transfected HEK 293T cells were reseeded on 12 mm round glass coverslips coated by poly-L-lysine. Experiments took place ~12–24 hr after transfection.

## Cell lines

HEK 293T and T-Rex 293 cell lines used in this study were respectively from the American Type Culture Collection and Thermo Fisher, authenticated by STR locus and tested negative for mycoplasma contamination.

## Mouse epidermal keratinocyte culture

The animal protocol used in this study was approved by the Institutional Animal Care and Use Committee of Wuhan University. Primary mouse keratinocytes were prepared according to the method previously described (*Luo et al., 2012*; *Pirrone et al., 2005*). Briefly, newborn WT C57B/6 mice (post-natal days 1–3) were deeply anaesthetized and decapitated and then soaked in 10% povidone-iodine, 70% ethanol, and phosphate-buffered saline (PBS) for 5 min, respectively. The skin on the back was removed and rinsed with pre-cold sterile PBS in a 100 mm Petri dish and transferred into a 2 ml tube filled with pre-cold digestion buffer containing 4 mg/ml dispase II and incubated overnight at 4°C. After treatment with dispase II for 12–18 hr, the epidermis was gently peeled off from dermis and collected. Keratinocytes were dispersed by gentle scraping and flushing with KC growth medium (Invitrogen). The resulting suspension of single cells was collected by centrifuge, and cells were seeded onto coverslips pre-coated with poly-L-lysine and maintained in complete keratinocyte serum-free growth medium (Invitrogen). Cell culture medium was refreshed every two days. Patch-clamp recordings were carried out 48 hr after plating.

## Electrophysiological recording

Conventional whole-cell and excised patch-clamp recording methods were used. For the recombinant expressing system, green fluorescent EGFP was used as a surface marker for gene expression. Recording pipettes were pulled from borosilicate glass capillaries (World Precision Instruments) and fire-polished to a resistance between 2 and 4 MΩ when filled with internal solution containing (in mM) 140 CsCl, 2.0 $MgCl_2$, 5 EGTA, and 10 HEPES, pH 7.4 (adjusted with CsOH). Bath solution contained (in mM): 140 NaCl, 5 KCl, 3 EGTA, and 10 HEPES, pH 7.4 adjusted with NaOH. For recordings in keratinocytes, the bath saline consisted of (in mM) 140 NaCl, 5 KCl, 2 $MgCl_2$, 2 $CaCl_2$, 10 glucoses, and 10 HEPES, pH 7.4 adjusted with NaOH, and the pipette solution contained (in mM) 140 CsCl, 5 EGTA, and 10 HEPES, pH 7.3 adjusted with CsOH. For single-channel recordings, the pipette solution and bath solution were symmetrical and contained (in mM) 140 NaCl, 5 KCl, 3 EGTA, and 10 HEPES, pH 7.4. Isolated cells were voltage clamped and held at –60 mV using an EPC10 amplifier with the Patchmaster software (HEKAElectronics, Lambrecht, Germany). For a subset of recordings, currents were amplified using an Axopatch 200B amplifier (Molecular Devices, Sunnyvale, CA) and recorded through a BNC-2090/MIO acquisition system (National Instruments, Austin, TX) using QStudio developed by Dr. Feng Qin at State University of New York at Buffalo. Whole-cell recordings were typically sampled at 5 kHz and filtered at 1 kHz, and single-channel recordings were sampled at 25 kHz and filtered at 10 kHz. The compensation of pipette series resistance and capacitance was compensated using the built-in circuitry of the amplifier (>80%) to reduce voltage errors. Exchange of external solution was performed using a gravity-driven local perfusion system. As determined by the conductance tests, the solution around a patch under study was fully controlled by the application of a flow rate of 100 µl/min or greater. Dyclonine hydrochloride, MTSET and carvacrol were purchased from MCE (MedChemExpress). Unless otherwise noted, all chemicals were purchased from Sigma (Millipore Sigma, St. Louis, MO). Water-insoluble reagents were dissolved in pure ethanol or DMSO to make a stock solution and diluted into the recording solution at the desired final concentrations before the experiment. The final concentrations of ethanol or DMSO did not exceed 0.3%, which had no effect to currents. In the scratching behavior experiments, carvacrol was first dissolved in 10% ethanol and then diluted in normal saline before administration. All experiments except those for heat activation were performed at room temperature (22–24°C).

## Ultrafast temperature jump achievement

Rapid temperature jumps were generated by the laser irradiation approach as described previously (*Yao et al., 2009*). In brief, a single-emitter infrared laser diode (1470 nm) was used as a heat source. A multimode fiber with a core diameter of 100 µm was used to transmit the launched laser beam. The other end of the fiber exposing the fiber core was placed close to cells as the perfusion pipette is

typically positioned. The laser diode was driven by a pulsed quasi-CW current powder supply (Stone Laser, Beijing, China). Pulsing of the controller was controlled from computer through the data acquisition card using QStudio software developed by Dr. Feng Qin at State University of New York at Buffalo. A blue laser line (460 nm) was coupled into the fiber to aid alignment. The beam spot on the coverslip was identified by illumination of GFP-expressing cells using the blue laser during experiment.

Constant temperature steps were generated by irradiating the tip of an open pipette and using the current of the electrode as the readout for feedback control. The laser was first powered on for a brief duration to reach the target temperature and then modulated to maintain a constant pipette current. The sequence of the modulation pulses was stored and subsequently played back to apply temperature jumps to the cell of interest. Temperature was calibrated offline from the pipette current using the dependence of electrolyte conductivity.

## Cell death analysis by flow cytometry

T-Rex 293 cells were grown in DMEM containing 4.5 mg/ml glucose, 10% (vol/vol) FBS, 50 units/ml penicillin, 50 μg/ml streptomycin, and blasticidin S (10 μg/ml), and were incubated at 37°C in a humidified incubator gassed with 5% $CO_2$. Transfections were performed in wells of a 24-well plate using lipofectamine 2000 (Invitrogen). The GFP-TRPV3 (WT and G573 mutants) cDNAs in pcDNA4/TO vector were individually transfected into T-Rex 293 cells and treated with 20 ng/ml doxycycline 16 hr post-transfection to induce the gene expression following the method as previously described (*Xiao et al., 2008*). Expression of GFP fluorescence detected by an epifluorescence microscope was used as an indicator of gene expression. After treatments with the compounds for 12 hr, cells were collected, washed twice with PBS, resuspended, and then dyed with propidium iodide (PI, Thermo Fisher Scientific) in the dark according to the manufacturer's instructions. The membrane integrity of the cells was assessed using a BD FACSCelesta flow cytometer equipped with BD Accuri C6 software (BD Biosciences, USA).

## Evaluation of scratching behavior in Mice

Behavioral studies were performed with 6- to 8-week-old WT or *Trpv3*$^{-/-}$ adult C57B/6 mice. To assess itch-scratching behaviors, the hair of the rostral part of the mouse's right neck was shaved using an electric hair clipper 24 hr before the start of experiments. *Trpv3*$^{-/-}$ mice have been described previously (*Wang et al., 2021*). Scratching behaviors were recorded on video. The number of itch-scratching bouts was counted through video playback analysis. One scratching bout was defined as an episode in which a mouse lifted its right hind limb to the injection site and scratched continuously for any time length until this limb was returned to the floor or mouth (*Wilson et al., 2013*). All behavioral experiments were conducted in a double-blind manner. To examine acute scratching/itch induced by carvacrol or pruritogen histamine, mice were first placed in an observation box (length, width, and height: 9 × 9 × 13 cm$^3$) for acclimatization for about 30 min. Then, carvacrol (0.1%) in a volume of 50 μl was injected intradermally into the right side of the mouse's neck. To access the effect of dyclonine on itch scratching, normal saline (0.9% NaCl) or dyclonine (1, 10, and 50 μM) was injected intradermally 30 min before intradermal injection of carvacrol (*Cui et al., 2018*; *Sun and Dong, 2016*). Behaviors were recorded on video for 30 min following the injection of carvacrol.

## Hargreaves test for behavioral experiments

All tests were conducted during the light phase of the light/dark cycle by a trained observer blind to the genotype. Mice were habituated to the testing room for 60 min prior to the behavioral tests unless otherwise stated. Hargreaves test was performed as described previously (*Wang et al., 2018*). All behavioral experiments were conducted in a double-blind manner. For measurement of thermal hyperalgesia, animals were placed individually, 30 min after injection, on a hot plate (Bioseb, Chaville, France) with the temperature adjusted to 55°C. The withdrawal latency of each hind paw was determined until nocifensive reaction appeared (licking foot). Right hind paws of mice were injected intraplantarly with 10 μl normal saline (0.9% NaCl). Left hind paws of mice were injected intraplantarly with 10 μl normal saline (supplemented with 10 or 50 μM dyclonine).

## Molecular docking

The molecular docking approach was used to model the interaction between dyclonine and TRPV3 channel protein (PDB ID code: 6DVZ) according to previous description (*Huang et al., 2014*; *Li et al., 2018*). The 3D structure of dyclonine was generated by LigPrep (*Gadakar et al., 2007*). Glide (*Friesner et al., 2004*) and IFD (*Sherman et al., 2006*) were employed to dock dyclonine into the potential binding. For Glide docking, the grid for the protein was defined as an enclosing cubic box within 34 Å to include the upper pore region and the central cavity of TRPV3, and the XP docking mode was selected. During in silico docking, at most 100,000 poses passed through for the initial phase of docking, of which the top 300 poses were processed with post-docking minimization. The threshold for rejecting minimized pose was set to 0.5 kcal/mol. A maximum of 200 poses were finally written out. The docking scores and dyclonine-residue interaction distance were summarized, sorted, and then plotted by Maestro. IFD was performed to refine the interaction between dyclonine and TRPV3 (*Sherman et al., 2006*), L655, I674 and G638 residues were chosen from the center of the docking box, respectively. During this docking process, the protein and the dyclonine were both flexible. All structural figures were made by PyMol (http://www.pymol.org).

## Statistics

Data were analyzed offline with Clampfit (Molecular Devices), IGOR (Wavemetrics, Lake Oswego, OR), SigmaPlot (SPSS Science, Chicago, IL), and OriginPro (OriginLab Corporation, MA). For concentration dependence analysis, the modified Hill equation was used: $Y = A1 + (A2 - A1)/(1 + (IC_{50}/[toxin])^{n_H})$, in which $IC_{50}$ is the half maximal effective concentration, and $n_H$ is the Hill coefficient. Unless stated otherwise, the data are expressed as mean ± standard error (SEM), from a population of cells (n), with statistical significance assessed by Student's *t*-test for two-group comparison or one-way analysis of variance (ANOVA) tests for multiple group comparisons. Significant difference is indicated by a p value less than 0.05 (*p<0.05, **p<0.01).

## Acknowledgements

We are grateful to our colleagues and members of Yao lab for comments and discussions. We would also like to thank the core facilities of College of Life Sciences at Wuhan University for technical help. This work was supported by grants from the National Natural Science Foundation of China (31830031, 31929003, 31871174, 31671209, and 31601864), Natural Science Foundation of Hubei Province (2017CFA063 and 2018CFA016), the Fundamental Research Funds for the Central Universities, the Natural Science Foundation of Jiangsu Province (BK20202002), Innovation and Entrepreneurship Talent Program of Jiangsu Province, and State Key Laboratory of Utilization of Woody Oil Resource with grant number 2019XK2002.

## Additional information

### Funding

| Funder | Grant reference number | Author |
|---|---|---|
| National Natural Science Foundation of China | 31830031 | Jing Yao |
| National Natural Science Foundation of China | 31929003 | Jing Yao |
| National Natural Science Foundation of China | 31871174 | Jing Yao |
| National Natural Science Foundation of China | 31671209 | Jing Yao |
| National Natural Science Foundation of China | 31601864 | Chang Xie |

| Funder | Grant reference number | Author |
|---|---|---|
| Natural Science Foundation of Hubei Province | 2017CFA063 | Jing Yao |
| Natural Science Foundation of Hubei Province | 2018CFA016 | Jing Yao |
| Natural Science Foundation of Jiangsu Province | BK20202002 | Ye Yu |
| Natural Science Foundation of Jiangsu Province | 2019XK2002 | Ye Yu |
| Fundamental Research Fund for the Central Universities | 2042021kf0218 | Jing Yao |

The funders had no role in study design, data collection and interpretation, or the decision to submit the work for publication.

## Author contributions

Qiang Liu, Data curation, Formal analysis, Investigation, Software, Validation, Writing – original draft, Writing – review and editing; Jin Wang, Data curation, Formal analysis, Investigation, Methodology, Software, Validation, Writing – original draft, Writing – review and editing; Xin Wei, Conghui Ping, Yue Gao, Peiyu Wang, Data curation, Formal analysis, Investigation, Methodology, Software, Validation; Juan Hu, Data curation, Formal analysis, Investigation, Methodology, Validation; Chang Xie, Data curation, Formal analysis, Investigation, Methodology, Software, Validation, Visualization; Peng Cao, Data curation, Formal analysis, Methodology, Resources, Software, Validation; Zhengyu Cao, Data curation, Formal analysis, Methodology, Resources, Validation, Visualization; Ye Yu, Data curation, Formal analysis, Methodology, Resources, Software, Validation, Writing – review and editing; Dongdong Li, Formal analysis, Methodology, Software, Validation, Visualization, Writing – review and editing; Jing Yao, Conceptualization, Data curation, Formal analysis, Funding acquisition, Investigation, Methodology, Project administration, Resources, Software, Supervision, Validation, Visualization, Writing – original draft, Writing – review and editing

## Author ORCIDs

Dongdong Li ![ORCID] http://orcid.org/0000-0002-6731-4771
Jing Yao ![ORCID] http://orcid.org/0000-0003-1844-3988

## Ethics

All mice were housed in the specific pathogen-free animal facility at Wuhan University and all animal experiments were in accordance with protocols were adhered to the Chinese National Laboratory Animal-Guideline for Ethical Review of Animal Welfare and approved by the Institutional Animal Care and Use Committee of Wuhan University (NO. WDSKY0201804). The mice were euthanized with $CO_2$ followed by various studies.

## Decision letter and Author response

Decision letter https://doi.org/10.7554/eLife.68128.sa1
Author response https://doi.org/10.7554/eLife.68128.sa2

# Additional files

## Supplementary files

• Transparent reporting form

## Data availability

All the data for Therapeutic inhibition of keratinocyte TRPV3 sensory channel by local anesthetic dyclonine have been deposited in Dyrad with DOI https://doi.org/10.5061/dryad.7d7wm37sq.

The following dataset was generated:

| Author(s) | Year | Dataset title | Dataset URL | Database and Identifier |
|---|---|---|---|---|
| Liu Q, Wang J, Wei X, Hu J, Ping C, Gao Y, Xie C, Wang P, Cao P, Cao Z, Yu Y, Li D, Yao J | 2021 | | https://doi.org/10.5061/dryad.7d7wm37sq | Dryad Digital Repository, 10.5061/dryad.7d7wm37sq |

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
