## [Decision Letter]

**Acceptance summary:**

TRPV3 is a non-selective cation channel that is important for skin physiology and hair growth, yet there are currently no specific drugs to either activate or inhibit this channel. Qiang Liu and collaborators discover that TRPV3 channels from mouse, human and frog are inhibited by the topical analgesic dyclonine regardless of whether the channel is activated by agonists, heat or voltage, with much higher potency than TRPV1, TRPV2 and TRPM8 channels. Dyclonine starkly reduces cell death that results from TRPV3 channel activation by either gain of function mutations known to cause skin pathologies as well as by direct activation of wild-type channels by agonist, and that application of dyclonine reduces scratching behavior in mice pre-treated with the TRPV3 channel agonist carvacrol. Using computational approaches together with mutagenesis, the authors provide evidence that dyclonine binds within the pore to inhibit the channel. This is a carefully done study with high-quality data that identifies a novel, reversible and relatively selective inhibitor for an important TRP channel for which few pharmacological tools exist.

**Decision letter after peer review:**

[Editors’ note: the authors submitted for reconsideration following the decision after peer review. What follows is the decision letter after the first round of review.]

Thank you for submitting your work entitled "Therapeutic inhibition of keratinocyte TRPV3 sensory channels by local anesthetic dyclonine" for consideration by *eLife*. Your article has been reviewed by 3 peer reviewers, and the evaluation has been overseen by a Reviewing Editor and a Senior Editor. The following individual involved in review of your submission has agreed to reveal their identity: Alexander Theodore Chesler (Reviewer #3).

Our decision has been reached after consultation between the reviewers. Based on these discussions and the individual reviews below, we regret to inform you that your work will not be considered further for publication in *eLife*.

The editors and reviewers enjoyed reading your manuscript and were enthusiastic about the potential of your findings. As you will see, all three reviewers have substantial concerns with your key conclusions regarding the mechanism of dyclonine inhibition of TRPV3, selectivity of the drug for TRPV3 and with the physiological roles of TRPV3. The reviewers have suggested additional experiments that could potentially address their concerns, however, doing so would require considerably more than the two months we typically allot for inviting revisions. However, given the potential significance of your study, we would be willing to consider a new manuscript that includes additional experiments that address all of the reviewer's concerns.

*Reviewer #2:*

The function of the TRPV3 non-selective cation channel is important for skin physiology and hair growth, and gain of function mutations in this channel cause human pathologies that are mirrored in animal models. There are currently no specific drugs to either activate or inhibit this channel. In this study by Qiang Liu and collaborators they identify that when activated by 2-APB, the TRPV3 channels from mouse, human and frog are inhibited by the topical analgesic dyclonine with much higher potency than TRPV1, TRPV2 and TRPM8 channels when these are activated by capsaicin, 2-APB and menthol, respectively. They also find that dyclonine inhibits mouse TRPV3 channels when activated by both voltage and heat. The authors also show that dyclonine starkly reduces cell death that results from TRPV3 channel activation by either gain of function mutations known to cause skin pathologies as well as by direct activation of wild-type channels by 2-APB, and that application of dyclonine reduces scratching behavior in mice pre-treated with the TRPV3 channel agonist carvacrol or by histamine. By performing single-channel recordings they show that dyclonine reduces the open probability of TRPV3 channels activated by 2-APB. Finally the authors identify a series of residues located in the pore cavity of the channel that when individually substituted by alanine either strongly reduce or increase the potency of channel inhibition by dyclonine, suggesting that the drug might bind in the pore to inhibit channel activity or to block cations from permeating through the open channel pore. This is a carefully done study with high-quality data that identifies a novel, reversible and relatively selective inhibitor for an important TRP channel for which few pharmacological tools exist. However, I think that some key considerations are missing regarding the mechanism of action of dyclonine on TRPV3 channels, as well as its specificity for this channel. I have the following specific concerns:

1. The authors should highlight the zero-current levels in all figure panels where current traces are displayed; this is critical to assess the amount of leak that is present in the recordings.

2. The authors have not addressed whether the inhibition of TRPV3 channels by dyclonine is state dependent. I think this is important for assessing the specificity of the drug on other channels as well as the quantitative effects on inhibition of the various examined TRPV3 channel mutants. The authors use a variety of 2-APB concentrations to activate TRPV3 channels across experiments, none of which are saturating, whereas near-maximal agonist concentrations are used for TRPV1 and TRPV2. In the case of the examined TRPV3 mutations, those that increase the apparent affinity for dyclonine exhibit currents that are much noisier and activate more slowly than wild-type, suggesting a negative effect of the mutation on gating, whereas mutations that reduce the apparent affinity have larger current densities, faster kinetics of activation and less noise once channel activation reaches steady-state, suggesting a positive effect on gating. If dyclonine inhibition were inversely proportional to open probability, the differences in apparent affinity observed between TRP channel subtypes or between TRPV3 mutants could thus potentially arise from differences in the open probability between channels at the conditions in which inhibition was measured for each. I think that the authors should provide data showing inhibition by dyclonine at super-saturating concentrations of 2-APB and show that it is comparable to inhibition at lower agonist concentrations.

3. I think that the inhibitory effect of dyclonine on TRPV1, TRPV2 and TRPM8 channels is not low enough as to say that it is a specific effect for TRPV3 channels – the EC50 for inhibition of human TRPV3 channels is only 2-fold lower than that for TRPV2 channels. The authors also only examined a small number of TRP channels. I consider that examining the effects of dyclonine on the TRPA1 channel is particularly important, as this channel is known to also contribute to the pathophysiology of itch. I think it would be better if the authors used more conservative language regarding specificity, stating simply that the apparent affinity under their experimental conditions is larger for TRPV3 than for those other TRP channels that were tested.

4. Regarding the clinical implications of the study, it would make it much stronger if the authors showed directly using electrophysiological recordings that dyclonine can inhibit the gain of function mutations that are known to cause disease (G573S and G573C).

5. The in vivo results are highly non-specific, and it is impossible to determine whether the effect of dyclonine that is observed is indeed mediated by TRPV3 channels. As described in the Introduction section, dyclonine was initially looked at by the authors because of its proved effect as topical anesthetic – it is thus not surprising that it reduces scratching behavior in mice. I think these results should be removed from the manuscript, or additional experiments using TRPV3-deficient animals should be included to show that the effect on behavior requires the presence of TRPV3.

6. I found it hard to rationalize the effects of the distinct TRPV3 mutations on inhibition based on the illustrations for the three proposed binding sites that are shown in Figure 8. The authors should distinguish between alternative hypothesis regarding the proximity of docked dyclonine to the various channel residues in each of the proposed binding sites, clearly highlighting those that are far from the bound drug or close to it in each case. They should also provide more information and a better rationale for their choice of structural model – why was that structure chosen? How were the distinct binding sites ranked? Importantly, the inner cavity of the pore is highly conserved in TRPV1, TRPV2 and TRPV3 channels, suggesting that pore blockers should affect these three channels similarly. Are the identified residues in the pore conserved in TRPV1 and TRPV2, and if they are, how do the authors explain the differences in inhibition observed between these three channels? The authors should provide a discussion that considers whether the effect of dyclonine is on cation conduction (pore block) or on gating (antagonism).

*Reviewer #3:*

Trpv3 is a non-selective cation channel that can be gated by temperature and environmental irritants. Inherited mutations in humans are linked to rare diseases that affect skin and gene perturbations in mice have shown the importance of this ion channel in maintaining the skin barrier and hair production. There is growing evidence that Trpv3 has a sensory role in temperature detection, pain and itch. For example, several natural products have been shown to activate the channel and produce the sensation of warming or the desire to scratch. To date, there are few antagonists, and these are not particularly selective for this receptor over other Trp channels. Thus, the discovery of selective Trpv3 antagonists would have clear clinical and experimental utilities.

The current study investigates how one specific compound, dyclonine, impacts Trpv3 gating. Dyclonine is a general anesthetic, mostly used orally for mouth pain and sore throats. Notably, the mechanism of action of this drug is unknown. Here, the authors have used electrical recordings, modeling and mutagenesis to provide evidence that dyclonine functions as a Trpv3 antagonist. In general, the data are straight forward and pretty convincing. Less clear is how potent and selective dyclonine really is and whether Trpv3 is the relevant target in vivo. Indeed, the in vivo experiments clearly indicate that this drug must have other targets. Specifically, the mechanism of histamine-evoked scratching is well understood and does not involved Trpv3. How then does dyclonine block this type of itch? To better understand what's going on, recordings from DRG neurons should be done in the presence and absence of dyclonine. Quantitation of neuronal excitability, biophysical properties and sensory responses to a range of agonists would be particularly informative. Second, behavioral tests need to be expanded to other modalities beyond itch (for which Trpv3 has a minor role). Warming and heat pain should be tested at the minimum.

*Reviewer #4:*

I limit my comments to the binding site analysis described in this article. There are several issues with this aspect of this manuscript.

1. The authors carried out docking to a cryo-EM structure of TRPV3 at 4.3Å resolution using Induced Fit Docking with the protein conformation fixed and the ligand flexible. At such a resolution the side chains in the pocket may well be very poorly defined and docking is unlikely to be reliable. It is not clear that the protein structure is suitable for docking, especially since the protein side chain orientations were fixed.

2. The center of the docking box was defined using L655, I674 and G638. The authors do not mention the box size that they use, but default box size in Induced Fit Docking, I believe, is 10x10x10 Å, which leaves very few alternatives for a ligand of this size. Therefore the residues reported to be found at the center of the binding pocket are, in essence, predefined (page 15).

3. The method used to select the three docking poses from the docking results is unclear. Were there only three results above a specific energy threshold? Did the authors do clustering of the results? More details are required.

4. The figures do not provide sufficient clarity to understand how the ligand is docking in the three poses mentioned. The overview (Figure 8A) is too general, while the close-up views (Figure 8 B) need labelling of the transmembrane helices and subunits.

5. While the mutagenesis provides interesting data about the residues that might be involved with the dyclonine effect, it could reflect indirect effects.

Taken together the presented computational and mutagenesis data should not be used as evidence that the binding site has been identified.

[Editors’ note: further revisions were suggested prior to acceptance, as described below.]

Thank you for resubmitting your work entitled "Therapeutic inhibition of keratinocyte TRPV3 sensory channel by local anesthetic dyclonine" for further consideration by *eLife*. Your revised article has been reviewed by 3 peer reviewers, and the evaluation has been overseen by Kenton Swartz as the Senior and Reviewing Editor.

The manuscript has been improved but there are some remaining issues that need to be addressed, as outlined below:

The reviewers believe the authors have done an excellent job of addressing most of the concerns raised in the last review of their manuscript, and by performing multiple complicated additional experiments they have provided solid support for most of their conclusions. The reviewers still have a few concerns that need to be addressed by careful revision of the manuscript.

1. We think the authors should tone down their statements throughout the manuscript about dyclonine functioning as a selective inhibitor of mouse and human TRPV3 channels, for the following reasons: First, as mentioned in the last review, the apparent affinity for dyclonine in TRPV2 (31 uM) is not that much lower than that of the human orthologue of TRPV3 when activated by 2-APB (16 uM) or of mouse TRPV3 activated by heat (14 uM), indicating that dyclonine might not be selective in humans. Second, the number of TRP channels tested is small, and no other channels were tested that could also contribute to the behavioral responses to carvacrol downstream of TRPV3 activation and that could also be affected by dyclonine. Third, TRPV3 is reportedly expressed in human DRG neurons (see Xu et al. Nature 2002 and Smith et al. Nature 2002) but not in mice DRGs (Peier et al. Science 2002), calling into question the relevance of the DRG data in Figure S1A-D to ascertain the specificity of dyclonine action on TRPV3. We think this should be discussed. Fourth, the differences in paw withdrawal latency between WT and TRPV3 KO mice with and without dyclonine (Figure S1E) are too small to be convincing, and there is significant overlap between the data points at all conditions. The authors should tone down their conclusions from these data and state that at best a minimal trend can be observed that suggests an effect of TRPV3 in the latency responses. Fifth, it would have been ideal to show that dyclonine has no effect on the 20% remaining response to carvacrol in TRPV3 KO mice in the scratching experiments in Figure 6. Otherwise with the data provided it can be concluded that TRPV3 is required for the majority of the response to carvacrol, but the data does not prove that TRPV3 is the sole target for the drug. It would also be ideal to for the authors to show that dyclonine also inhibits TRPV3 channel activation by carvacrol. Finally, we don't understand how dyclonine can be a general anesthetic (in people) if it is functioning solely through Trpv3. This point should be discussed and the authors should acknowledge that the drug may have other targets.

2. The sensitization experiments in Figure 1G-I are not of adequate quality to conclude that dyclonine slows sensitization of the channel; the peak current magnitudes in the last stimulation with 2-APB in Figures 1G and 1H are noticeably larger than the second-to last peak current magnitudes. This suggests that sensitization has not yet reached equilibrium in either case, and yet because experiments without dyclonine were shorter than experiments with dyclonine, it is difficult to determine whether the two are indeed different. We think the authors should show the same data normalized to the initial stimulation with 2-APB instead of normalizing to the last response. This would provide a much clearer way of comparing the two time-courses given that equilibrium has not been reached. We realize that this point does not alter the main conclusions in the paper, but data should be analyzed and discussed in the most accurate way possible regardless.

3. In relation to the absence of voltage-dependence of inhibition by dyclonine, it is hard to reconcile the data in Figure 3C with the current-voltage relations shown in the upper panel – at negative membrane potentials a concentration of 30 μm dyclonine certainly seems to inhibit as much current as ruthenium red (RR), which we assume provides the baseline for maximal inhibition and would thus represent 100% inhibition instead of 50% as indicated in the lower panel. The authors need to include data for the baseline in the absence of agonists to accurately assess the level of inhibition. In addition, the voltage dependence of RR should be predictably larger than that of dyclonine because it is a hexavalent cation. We think it is necessary for the authors to clarify these discrepancies in the data in order to strongly conclude that there is no voltage dependence to dyclonine action.

4. In relation to the data in Figure 5E, the fluorescence intensity in GFP expression is not an accurate way to estimate protein expression in general, and even if GFP were covalently attached to the channel, it would still be difficult to estimate channel expression from the intensity of GFP because TRP channels tend to accumulate in intracellular compartments when over expressed in heterologous systems. The authors should remove all statements regarding protein expression levels.

5. There is indeed a clear decrease in affinity for dyclonine in TRPV1 and TRPV2 channels compared to mouse TRPV3 channels activated by 2-APB. We think providing additional discussion about the sequence and structural differences between the three channels near the proposed binding site for dyclonine would be interesting for readers and might provide additional interesting insight into the potential underlying mechanism of inhibition.

6. We think the authors have done a nice job of providing more information for the docking analysis and we think this provides valuable information for where dyclonine may bind. Having said that, we also have serious concerns regarding the new metadynamics calculations, and to avoid delaying publication further, would recommend removing them, as the docking data is sufficient basis for the mutagenesis that tests the site.

First, the methods for the metadynamics include several references to previously-published simulations on TRPV3 and P2X3. It's not clear which parts are taken from which paper. For example, was the well-tempered variant of metadynamics used here? Second, and most importantly, the choice and definitions of the collective variables are unclear. Specifically, why do these definitions represent the possible binding modes of the drug and how can we be sure that they are not biased to what was observed in the docking? Where is E631 (not shown in any figures, nor described) and why was its distance to the N in the ring of dyclonine used for CV1? Similarly, what are the "colored carbon atoms of dyclonine" used to define the dihedral angle in CV2? (Perhaps the authors mean the oxygen and nitrogen atoms, but I count only three atoms of this type, while a dihedral angle requires four atoms. It doesn't help that the (presumably red) oxygen atoms are hard to differentiate from the orange of the carbon atoms in Figure 8A and B). Third, the figure of the results (Figure 8B) indicates two configurations with energy minima at a distance of 9 Å (BM_A1_ and BM_A2_). However, it seems to us that those minima at +3 and -3 radians are actually related, given the continuity of angle space. Also, the axis labels are almost impossible to read on the energy landscape plot. Fourth, we are confused by the description of site BM(A): page 17, line 15, says it is formed between the pore loop and S5, whereas Figure 8A/B shows it between the pore helix and S6. Is this a typo?

---

## [Author Response]

Reviewer #2:

The function of the TRPV3 non-selective cation channel is important for skin physiology and hair growth, and gain of function mutations in this channel cause human pathologies that are mirrored in animal models. There are currently no specific drugs to either activate or inhibit this channel. In this study by Qiang Liu and collaborators they identify that when activated by 2-APB, the TRPV3 channels from mouse, human and frog are inhibited by the topical analgesic dyclonine with much higher potency than TRPV1, TRPV2 and TRPM8 channels when these are activated by capsaicin, 2-APB and menthol, respectively. They also find that dyclonine inhibits mouse TRPV3 channels when activated by both voltage and heat. The authors also show that dyclonine starkly reduces cell death that results from TRPV3 channel activation by either gain of function mutations known to cause skin pathologies as well as by direct activation of wild-type channels by 2-APB, and that application of dyclonine reduces scratching behavior in mice pre-treated with the TRPV3 channel agonist carvacrol or by histamine. By performing single-channel recordings they show that dyclonine reduces the open probability of TRPV3 channels activated by 2-APB. Finally the authors identify a series of residues located in the pore cavity of the channel that when individually substituted by alanine either strongly reduce or increase the potency of channel inhibition by dyclonine, suggesting that the drug might bind in the pore to inhibit channel activity or to block cations from permeating through the open channel pore. This is a carefully done study with high-quality data that identifies a novel, reversible and relatively selective inhibitor for an important TRP channel for which few pharmacological tools exist. However, I think that some key considerations are missing regarding the mechanism of action of dyclonine on TRPV3 channels, as well as its specificity for this channel. I have the following specific concerns:

1. The authors should highlight the zero-current levels in all figure panels where current traces are displayed; this is critical to assess the amount of leak that is present in the recordings.

We have now added the zero-current levels, shown as the dotted lines, to all current traces.

2. The authors have not addressed whether the inhibition of TRPV3 channels by dyclonine is state dependent. I think this is important for assessing the specificity of the drug on other channels as well as the quantitative effects on inhibition of the various examined TRPV3 channel mutants. The authors use a variety of 2-APB concentrations to activate TRPV3 channels across experiments, none of which are saturating, whereas near-maximal agonist concentrations are used for TRPV1 and TRPV2. In the case of the examined TRPV3 mutations, those that increase the apparent affinity for dyclonine exhibit currents that are much noisier and activate more slowly than wild-type, suggesting a negative effect of the mutation on gating, whereas mutations that reduce the apparent affinity have larger current densities, faster kinetics of activation and less noise once channel activation reaches steady-state, suggesting a positive effect on gating. If dyclonine inhibition were inversely proportional to open probability, the differences in apparent affinity observed between TRP channel subtypes or between TRPV3 mutants could thus potentially arise from differences in the open probability between channels at the conditions in which inhibition was measured for each. I think that the authors should provide data showing inhibition by dyclonine at super-saturating concentrations of 2-APB and show that it is comparable to inhibition at lower agonist concentrations.

As suggested by the reviewer, we have performed new experiments to address this issue: (1) As shown in Figure 1D-E, the inhibitory effect of dyclonine on TRPV3 is comparable under different concentration of 2-APB. These results also indicate that 300 µM 2-APB reaches the saturating activation level; (2) We further validated the inhibitory effect of dyclonine on TRPV3 when it was activated by super-saturating concentration of 2-APB (1 mM) in Figure 2A_3_ and Figure 8C (L642A, I659A mutations). These results do strengthen our conclusion that dyclonine potently inhibits TRPV3 activity. The new data are now described from page 7-8**.**

3. I think that the inhibitory effect of dyclonine on TRPV1, TRPV2 and TRPM8 channels is not low enough as to say that it is a specific effect for TRPV3 channels – the EC50 for inhibition of human TRPV3 channels is only 2-fold lower than that for TRPV2 channels. The authors also only examined a small number of TRP channels. I consider that examining the effects of dyclonine on the TRPA1 channel is particularly important, as this channel is known to also contribute to the pathophysiology of itch. I think it would be better if the authors used more conservative language regarding specificity, stating simply that the apparent affinity under their experimental conditions is larger for TRPV3 than for those other TRP channels that were tested.

The x-axis in Figure 2C is logarithmic. The IC50 of dyclonine inhibition for TRPV3 is in fact about 10-fold lower than TRPV2 (Figure 2C), 20-fold lower than TRPM8, and 100-fold lower than TRPV1. We are sorry for not clearly conveying this information.

As suggested by the reviewer, we also performed new experiments to examine the effect of dyclonine on TRPA1 channel that was activated by the specific agonist AITC (Allyl Isothiocyanate). The results are now shown in Figure 2A_5_, 2B and 2C. The IC50 of dyclonine for TRPA1 is about 50-fold higher than for TRPV3 (Figure 2C), supporting dyclonine’s specific inhibition for TRPV3 channel. We describe the new data now on page 9 2^nd^ paragraph.

4. Regarding the clinical implications of the study, it would make it much stronger if the authors showed directly using electrophysiological recordings that dyclonine can inhibit the gain of function mutations that are known to cause disease (G573S and G573C).

We performed new experiments to validate the inhibitory effect of dyclonine on the diseases-related G573S and G573C TRPV3 mutants. Our new results show that TRPV3 mutants display gain-of-function basal currents which were transiently blocked by dyclonine application and the wide-spectra TRP channel blocker ruthenium red (RR) (data added now to Figure 5A, 5B and 5C). Moreover, we did experiments to show that for the two TRPV3 mutants, dyclonine also inhibited the evoked currents by saturating concentration of 2-APB (300 µM; new data in Figure 5A, 5B and 5D). We describe the results now on page 12.

5. The in vivo results are highly non-specific, and it is impossible to determine whether the effect of dyclonine that is observed is indeed mediated by TRPV3 channels. As described in the Introduction section, dyclonine was initially looked at by the authors because of its proved effect as topical anesthetic – it is thus not surprising that it reduces scratching behavior in mice. I think these results should be removed from the manuscript, or additional experiments using TRPV3-deficient animals should be included to show that the effect on behavior requires the presence of TRPV3.

We performed new experiments using TRPV3 knock out mice, to evaluate the contribution of TRPV3 to the observed behavior. We observed that TRPV3 KO largely reduced carvacrol-caused scratching behavior (~80% reduction), indicating that TRPV3 underlies the scratching responses (new data now in Figure 6A-B). The inhibition effect of dyclonine on carvacrol-caused scratching (Figure 6C-D) thus involves the specific inhibition of TRPV3 channel. Also, we have removed the results on histamine-evoked scratching behavior since the scratch/itch behavior caused by histamine is more complicated and whether the TRPV3 channel contributes to it is still debated. These new results are described on page 14.

6. I found it hard to rationalize the effects of the distinct TRPV3 mutations on inhibition based on the illustrations for the three proposed binding sites that are shown in Figure 8. The authors should distinguish between alternative hypothesis regarding the proximity of docked dyclonine to the various channel residues in each of the proposed binding sites, clearly highlighting those that are far from the bound drug or close to it in each case. They should also provide more information and a better rationale for their choice of structural model – why was that structure chosen? How were the distinct binding sites ranked? Importantly, the inner cavity of the pore is highly conserved in TRPV1, TRPV2 and TRPV3 channels, suggesting that pore blockers should affect these three channels similarly. Are the identified residues in the pore conserved in TRPV1 and TRPV2, and if they are, how do the authors explain the differences in inhibition observed between these three channels? The authors should provide a discussion that considers whether the effect of dyclonine is on cation conduction (pore block) or on gating (antagonism).

We have thoroughly revised this Results section (‘The mechanism underlying the inhibition of TRPV3 by dyclonine’)*,* and extend the discussion following the reviewer’s suggestion (page 21, 2^nd^ paragraph to end).

We performed new simulation analysis, by taking into account the relative proximity of docked dyclonine to the identified residues using extra precision scoring (XP), Induced-Fit-Docking (IFD) and metadynamics algorithm (page 16 – 17). There are three possible binding modes revealed by our new simulations, and we carefully compared their relevance on dyclonine inhibition. Our analysis suggests that the binding mode BM_A_ likely accounts for the dyclonine-TRPV3 interaction, which was further verified through single-residue mutation and electrophysiological experiments.

Regarding the cavity of channel pores of other TRPV channels, although similarity exits, the amino acids that make up the pocket and the precise shape of the pocket are diverse (Figure 8—figure supplement 2). This may be the reason why TRPV3 is selectively targeted by dyclonine. We now mention this point in the discussion on page 21 lines 20-22.

Reviewer #3:

Trpv3 is a non-selective cation channel that can be gated by temperature and environmental irritants. Inherited mutations in humans are linked to rare diseases that affect skin and gene perturbations in mice have shown the importance of this ion channel in maintaining the skin barrier and hair production. There is growing evidence that Trpv3 has a sensory role in temperature detection, pain and itch. For example, several natural products have been shown to activate the channel and produce the sensation of warming or the desire to scratch. To date, there are few antagonists, and these are not particularly selective for this receptor over other Trp channels. Thus, the discovery of selective Trpv3 antagonists would have clear clinical and experimental utilities.

The current study investigates how one specific compound, dyclonine, impacts Trpv3 gating. Dyclonine is a general anesthetic, mostly used orally for mouth pain and sore throats. Notably, the mechanism of action of this drug is unknown. Here, the authors have used electrical recordings, modeling and mutagenesis to provide evidence that dyclonine functions as a Trpv3 antagonist. In general, the data are straight forward and pretty convincing. Less clear is how potent and selective dyclonine really is and whether Trpv3 is the relevant target in vivo. Indeed, the in vivo experiments clearly indicate that this drug must have other targets. Specifically, the mechanism of histamine-evoked scratching is well understood and does not involved Trpv3. How then does dyclonine block this type of itch? To better understand what's going on, recordings from DRG neurons should be done in the presence and absence of dyclonine. Quantitation of neuronal excitability, biophysical properties and sensory responses to a range of agonists would be particularly informative. Second, behavioral tests need to be expanded to other modalities beyond itch (for which Trpv3 has a minor role). Warming and heat pain should be tested at the minimum.

1. Although there are some literatures about histamine-induced itching mediated by TRPV3 activation (Asakawa et al., 2006; Sun et al., 2018; Zhang et al., 2019), this issue remains debated. We agree with the reviewer’s concern, and we have now removed the in vivo data from histamine treatment. To validate the involvement of TRPV3 in carvacrol-caused scratching behavior, we performed new experiments using TRPV3 knock out (KO) mice. We observed that TRPV3 KO largely reduced carvacrol-caused scratching (~80% reduction), indicating that TRPV3 underlies the scratching response (new data on Figure 6A-B). Hence, the inhibition effect of dyclonine on carvacrol-caused scratching involves the specific inhibition of TRPV3 channel (Figure 6C-D). These results are described on page 14 lines 2-13.

2. Also following the suggestion of the reviewer, we performed new electrophysiological experiments in dorsal root ganglia (DRG) and trigeminal ganglia (TG) neurons, in the presence and absence of dyclonine. We found that the resting membrane potential of both types of neurons was unaffected by the presence of 10 µM (new data now in supplementary Figure 1). These results suggest the specific targeting of keratinocyte TRPV3 by dyclonine. This point is described from page 14 line 20 to page 15 line 2.

3. In addition to the itching/scratching behavior, we did new experiments in wild-type and TRPV3 KO mice to examine the effect of dyclonine on thermal nociceptive responses to the noxious temperature 55 °C. The results indicate that in wild-type mice, dyclonine dose-dependently reduced the nociceptive response (new data in supplementary Figure 1). TRPV3 KO reduced mice nociceptive response to heating (55°C; comparison between grey bars in supplementary Figure 1E). However, in TRPV3 KO mice, dyclonine showed no further effect, indicating that dyclonine specifically targets TRPV3 in vivo. These new data also suggest that TRPV3 partially contributes to pain sensation in thermal nociception, in line with its polymodal functions in cellular sensing. We describe these results on page 15 lines 3-11.

Reviewer #4:

I limit my comments to the binding site analysis described in this article. There are several issues with this aspect of this manuscript.

We have thoroughly revised this Results section (‘The mechanism underlying the inhibition of TRPV3 by dyclonine’, page 16-19). Several new sets of simulation analysis and molecular mutation experiments were performed. The updated results suggest that dyclonine interacts with the pore cavity of TRPV3 to exert the inhibition.

1. The authors carried out docking to a cryo-EM structure of TRPV3 at 4.3Å resolution using Induced Fit Docking with the protein conformation fixed and the ligand flexible. At such a resolution the side chains in the pocket may well be very poorly defined and docking is unlikely to be reliable. It is not clear that the protein structure is suitable for docking, especially since the protein side chain orientations were fixed.

The induced-fit-docking (IFD) was performed with a configuration where both the ligand and residue side chains of TRPV3 within 10Ả of the ligand pose are flexible.

We agree with the reviewer that the cryo-EM structure of TRPV3 at 4.3Å resolution is not optimal for *in silico* docking. To address this, the free energy profiles for TRPV3/dyclonine interaction were further reconstructed by metadynamics, a powerful algorithm for free energy reconstruction in complex Hamiltonians’ systems (Laio and Gervasio, 2008). Then, the TRPV3-dyclonine contacts were extensively sampled (Figure S2A-C). We identified a dyclonine pose with the lowest free energy as the potential binding mode to TRPV3, which is almost identical to the mode BM_A_ derived by IFD (Figure 8A-B), suggesting that the IFD model is reasonable, in spite of the relatively low resolution of the mTRPV3 cryo-EM structure. We describe these new analysis now from page 16 to page 18.

We further demonstrated the functional relevance of the residues located in BM_A_ binding pocket in TRPV3 activation and in dyclonine inhibition, using covalent occupation at this cavity (new data now in Figure 8G-I). Covalent modifications of L630C, F633C and L642C using MTSET (2-(trimethylammonium) ethyl methanethiosulfonate, bromide) significantly decreased 2-APB-idnuced current of these mutated mTRPV3 channels, while the reduction regent DTT rescued the current, indicating that the interruption of the allostery of the pore cavity has impaired the channel activation. In contrast, MSTET’s treatment had no effect on the activation of wild-type TRPV3 channel.

Corresponding to above observation, the MTSET-modification also reduced the relative inhibition effect (*i.e.,* comparing 2-APB-evoked currents in the absence and presence of dyclonine per condition) of dyclonine in L630C, F633C and L642C mutants, but not in wild-type TRPV3 channel. DTT could restore the relative inhibition effect of dyclonine in these mutants.

These new analysis and experimental results demonstrate the functional involvement of BM_A_ binding pocket in dyclonine-mediated TRPV3 inhibition. These new data are described from page 18 to page 19.

2. The center of the docking box was defined using L655, I674 and G638. The authors do not mention the box size that they use, but default box size in Induced Fit Docking, I believe, is 10x10x10 Å, which leaves very few alternatives for a ligand of this size. Therefore the residues reported to be found at the center of the binding pocket are, in essence, predefined (page 15).

Yes, in induced-fit-docking, residues within 10 Å of the dyclonine pose were extensively optimized. The IFD model used here is aimed to optimize the dyclonine pose.

The binding modes screening in the pore region was further carried out using extra precision (XP) scoring of GLIDE, in which the box size was set as 34 Å. This information was missed in the original manuscript, and now incorporated into page 16 lines 20 – 22.

3. The method used to select the three docking poses from the docking results is unclear. Were there only three results above a specific energy threshold? Did the authors do clustering of the results? More details are required.

We are sorry for missing this information. Yes, the results were clustered in the XP scoring (Figure S2B, C). Details about the method used to select and define the three docking poses from the docking results are now substantially updated, and provided on page 16 2^nd^ paragraph to page 17.

Briefly, dyclonine inhibited TRPV3 currents evoked by either 2-APB or heating, implying that dyclonine is not a competitive antagonist (Figure 1 and 4). We therefore suspected that the inhibition effect of dyclonine arises from its interaction with the aqueous pore of open TRPV3. The grid file of *in silico* docking was constructed to mainly examine residues in the upper pore region and the central cavity of TRPV3 (Figure S2); then, the best receptor–ligand complex was evaluated by the extra precision (XP) scoring. Ligand clusters derived from XP docking suggested three TRPV3/ dyclonine binding modes (BMs): BM_A_, BM_B_ and BM_C_ (Figure S2A-C). Moreover, residues within 10 Å of dyclonine poses were extensively refined using Induced-Fit-Docking (IFD). BM_B_ and BM_C_ modes predict that dyclonine occupies the ion permeation pathway behaving as an open channel blocker. However, dyclonine is a positive charged alkaloid (Figure 8B) and its inhibition effect is voltage-independent (Figure 3), which contradicts with the open channel blocker prediction hence makes BM_B_ and BM_C_ unlikely.

Further analysis by metadynamics algorithm supports BM_A_ binding mode for dyclonine-TRPV3 interaction, where dyclonine makes contacts with the cavity formed by the pore loop and S5-helix of TRPV3 (Figure 8A and Figure S2B). This binding mode yields the lowest free energy (Figure 8B, BM_A1_). The superimposition of *apo* and open structures of TRPV3 revealed remarkable allosteric changes and cavity size reduction in these regions (Figure S2I, J), supporting the rationality of the BM_A_ mode. Hence, these new analysis imply that dyclonine likely behaves as a negative allosteric modulator.

4. The figures do not provide sufficient clarity to understand how the ligand is docking in the three poses mentioned. The overview (Figure 8A) is too general, while the close-up views (Figure 8 B) need labelling of the transmembrane helices and subunits.

Following this suggestion, we have updated these panels in the revised manuscript (Figure 8A, B and Figure S2A, B, D). The labelling of the transmembrane helices and subunits are added.

5. While the mutagenesis provides interesting data about the residues that might be involved with the dyclonine effect, it could reflect indirect effects.

We provide several lines of data suggesting that dyclonine interacts with the pore cavity residues of TRPV3 and operates likely as a negative allosteric regulator. That being said, we agree with the reviewer that the inhibition caused by dyclonine may be mediated by possible indirect effects with other residues. We page 22 lines 1-7). highlight this issue in the end of discussion, while leaving open for future investigations (

Taken together the presented computational and mutagenesis data should not be used as evidence that the binding site has been identified.

We agree; we now conservatively state and discuss our results (last paragraph of discussion). To conclude, in this study, we provide comprehensive data demonstrating the effective inhibition of TRPV3 by the licensed drug dyclonine, and our current data pave the way for identifying the exact dyclonine binding site in future investigations.

**References:**

Asakawa, M., Yoshioka, T., Matsutani, T., Hikita, I., Suzuki, M., Oshima, I.,... Sakata, T. (2006). Association of a mutation in TRPV3 with defective hair growth in rodents. J Invest Dermatol, 126(12), 2664-2672. doi:10.1038/sj.jid.5700468

Sun, X. Y., Sun, L. L., Qi, H., Gao, Q., Wang, G. X., Wei, N. N., and Wang, K. (2018). Antipruritic Effect of Natural Coumarin Osthole through Selective Inhibition of

Thermosensitive TRPV3 Channel in the Skin. Mol Pharmacol, 94(4), 1164-

1173. doi:10.1124/mol.118.112466

Zhang, H., Sun, X., Qi, H., Ma, Q., Zhou, Q., Wang, W., and Wang, K. (2019).

Pharmacological Inhibition of the Temperature-Sensitive and Ca(2+)Permeable Transient Receptor Potential Vanilloid TRPV3 Channel by Natural Forsythoside B Attenuates Pruritus and Cytotoxicity of Keratinocytes. J Pharmacol Exp Ther, 368(1), 21-31. doi:10.1124/jpet.118.254045

Laio, A., and Gervasio, F. L. (2008). Metadynamics: a method to simulate rare events and reconstruct the free energy in biophysics, chemistry and material science. Reports on Progress in Physics, 71(12). doi:Artn 12660110.1088/00344885/71/12/126601

[Editors’ note: further revisions were suggested prior to acceptance, as described below.]

The manuscript has been improved but there are some remaining issues that need to be addressed, as outlined below:

The reviewers believe the authors have done an excellent job of addressing most of the concerns raised in the last review of their manuscript, and by performing multiple complicated additional experiments they have provided solid support for most of their conclusions. The reviewers still have a few concerns that need to be addressed by careful revision of the manuscript.

1. We think the authors should tone down their statements throughout the manuscript about dyclonine functioning as a selective inhibitor of mouse and human TRPV3 channels, for the following reasons:

Reply: In general, we agree with the reviewers’ suggestion and have toned down the statements on the selective effects of dycloine by removing ‘selective’ through out the manuscript, while conservatively describing the effective inhibition on TRPV3 channel. Updates are made on P3, line 7; P5, line 13 ; P6, line 4 and line 11; P9, line 4 and line 17; P10, line 3; P13, line 19; P20, line 12 and line 19; P21, line 10; P22, line 4.

First, as mentioned in the last review, the apparent affinity for dyclonine in TRPV2 (31 uM) is not that much lower than that of the human orthologue of TRPV3 when activated by 2-APB (16 uM) or of mouse TRPV3 activated by heat (14 uM), indicating that dyclonine might not be selective in humans. Second, the number of TRP channels tested is small, and no other channels were tested that could also contribute to the behavioral responses to carvacrol downstream of TRPV3 activation and that could also be affected by dyclonine.

We agree with these two concerns.

Third, TRPV3 is reportedly expressed in human DRG neurons (see Xu et al. Nature 2002 and Smith et al. Nature 2002) but not in mice DRGs (Peier et al. Science 2002), calling into question the relevance of the DRG data in Figure S1A-D to ascertain the specificity of dyclonine action on TRPV3. We think this should be discussed.

Following the suggestion of the previous review, we tested the effects of dyclonine on neuron excitability in mouse DRG and trigeminal ganglia (TG) neurons. The results showed no effect of dyclonine (Figure 6—figure supplement 1A-D), which do corroborate the absence of the expression of TRPV3 in mouse DRG. These data also suggest that the in vivo effect of dyclonine arises from the targeting of keratinocyte TRPV3. We add this comment on P15, lines 6 to 9.

Fourth, the differences in paw withdrawal latency between WT and TRPV3 KO mice with and without dyclonine (Figure S1E) are too small to be convincing, and there is significant overlap between the data points at all conditions. The authors should tone down their conclusions from these data and state that at best a minimal trend can be observed that suggests an effect of TRPV3 in the latency responses.

For the description of thermal nociception, we have modified it as ‘In wild-type mice, dyclonine exhibited a tendency to reduce the nociceptive response’ on P15, line 12. Moreover, we replaced ‘specifically’with ‘mainly’ on P15, line 16.

Fifth, it would have been ideal to show that dyclonine has no effect on the 20% remaining response to carvacrol in TRPV3 KO mice in the scratching experiments in Figure 6. Otherwise with the data provided it can be concluded that TRPV3 is required for the majority of the response to carvacrol, but the data does not prove that TRPV3 is the sole target for the drug. It would also be ideal to for the authors to show that dyclonine also inhibits TRPV3 channel activation by carvacrol. Finally, we don't understand how dyclonine can be a general anesthetic (in people) if it is functioning solely through Trpv3. This point should be discussed and the authors should acknowledge that the drug may have other targets.

According to the suggestion, we performed new whole-cell recordings in TRPV3-expressing HEK 293T cells to further confirm the inhibitory effect of dyclonine on TRPV3 currents activated by carvacrol. Similar to that observed with the inhibition of 2-APB-evoked TRPV3 currents (Figure 1A-C), dyclonine also inhibited carvacrol-activated TRPV3 currents in a concentration-dependent manner with IC_50_ = 3.5 ± 0.34 M following sensitization by repeated application of 0.3 mM 2-APB (*n* = 8, new Figure 6E-F), implying that the itching caused by carvacrol is mainly due to the activation of TRPV3. We have now added this in the result section on P14, line 19 to P15, line 4.

We completely agree with the comments that dyclonine as a general anesthetic may have other targets. For example, voltage-gated sodium channels have been suggested to be inhibited by local anesthetic. We have now added this note into the discussion part on P21, lines 11-15.

2. The sensitization experiments in Figure 1G-I are not of adequate quality to conclude that dyclonine slows sensitization of the channel; the peak current magnitudes in the last stimulation with 2-APB in Figures 1G and 1H are noticeably larger than the second-to last peak current magnitudes. This suggests that sensitization has not yet reached equilibrium in either case, and yet because experiments without dyclonine were shorter than experiments with dyclonine, it is difficult to determine whether the two are indeed different. We think the authors should show the same data normalized to the initial stimulation with 2-APB instead of normalizing to the last response. This would provide a much clearer way of comparing the two time-courses given that equilibrium has not been reached. We realize that this point does not alter the main conclusions in the paper, but data should be analyzed and discussed in the most accurate way possible regardless.

Thanks for the careful reading and helpful suggestions. We re-analyzed the sensentization process by normalizing all responses to its initial response, and made a new plot as shown in Figure 1I.

3. In relation to the absence of voltage-dependence of inhibition by dyclonine, it is hard to reconcile the data in Figure 3C with the current-voltage relations shown in the upper panel – at negative membrane potentials a concentration of 30 μm dyclonine certainly seems to inhibit as much current as ruthenium red (RR), which we assume provides the baseline for maximal inhibition and would thus represent 100% inhibition instead of 50% as indicated in the lower panel. The authors need to include data for the baseline in the absence of agonists to accurately assess the level of inhibition. In addition, the voltage dependence of RR should be predictably larger than that of dyclonine because it is a hexavalent cation. We think it is necessary for the authors to clarify these discrepancies in the data in order to strongly conclude that there is no voltage dependence to dyclonine action.

We apologize for our carlessness. We agree with the reviewers’ judgment. As per your suggestions, we added the baseline in the absence of agonists to Figure 3B and recalculated the inhibition effects of dyclonine on TRPV3 by subtracting leak currents under different voltages (new Figure 3C). The result shows an enhanced inhibition efficiency of dyclonine, with the voltage-independence being retained.

4. In relation to the data in Figure 5E, the fluorescence intensity in GFP expression is not an accurate way to estimate protein expression in general, and even if GFP were covalently attached to the channel, it would still be difficult to estimate channel expression from the intensity of GFP because TRP channels tend to accumulate in intracellular compartments when over expressed in heterologous systems. The authors should remove all statements regarding protein expression levels.

We appreciate this constructive comment. We have deleted all statements regarding protein expression levels in the manuscript (P12, line 22 and P13, line 4).

5. There is indeed a clear decrease in affinity for dyclonine in TRPV1 and TRPV2 channels compared to mouse TRPV3 channels activated by 2-APB. We think providing additional discussion about the sequence and structural differences between the three channels near the proposed binding site for dyclonine would be interesting for readers and might provide additional interesting insight into the potential underlying mechanism of inhibition.

Thanks for the suggestions. We have now incorporated the sequence alignment of the pore region among the three channels into new Figure 8—figure supplement 2, and discussed the possible reasons for the different inhibitory effects of dyclonine on TRPV3 from TRPV1 and TRPV2 channels, on P22, lines 2-5.

6. We think the authors have done a nice job of providing more information for the docking analysis and we think this provides valuable information for where dyclonine may bind. Having said that, we also have serious concerns regarding the new metadynamics calculations, and to avoid delaying publication further, would recommend removing them, as the docking data is sufficient basis for the mutagenesis that tests the site.

First, the methods for the metadynamics include several references to previously-published simulations on TRPV3 and P2X3. It's not clear which parts are taken from which paper. For example, was the well-tempered variant of metadynamics used here? Second, and most importantly, the choice and definitions of the collective variables are unclear. Specifically, why do these definitions represent the possible binding modes of the drug and how can we be sure that they are not biased to what was observed in the docking? Where is E631 (not shown in any figures, nor described) and why was its distance to the N in the ring of dyclonine used for CV1? Similarly, what are the "colored carbon atoms of dyclonine" used to define the dihedral angle in CV2? (Perhaps the authors mean the oxygen and nitrogen atoms, but I count only three atoms of this type, while a dihedral angle requires four atoms. It doesn't help that the (presumably red) oxygen atoms are hard to differentiate from the orange of the carbon atoms in Figure 8A and B). Third, the figure of the results (Figure 8B) indicates two configurations with energy minima at a distance of 9 Å (BM_A1_ and BM_A2_). However, it seems to us that those minima at +3 and -3 radians are actually related, given the continuity of angle space. Also, the axis labels are almost impossible to read on the energy landscape plot. Fourth, we are confused by the description of site BM(A): page 17, line 15, says it is formed between the pore loop and S5, whereas Figure 8A/B shows it between the pore helix and S6. Is this a typo?

We agree with the reviewers’ suggestion, and have removed metadynamics calculations in the revised manuscript. We also corrrected the description of site BM(A) as ‘BM_A_ mode shows that dyclonine makes contacts with the cavity formed by the pore loop and S6-helix of TRPV3’ on P17, from line 20.